# Solution Augmentation for ARC-AGI Problems Using GFlowNet: A Probabilistic Exploration Approach

**Sanha Hwang**                                         *hsh6449@gmail.com*
*Department of AI Convergence*
*Gwangju Institute of Science and Technology*

**Seungpil Lee**                                     *iamseungpil@gm.gist.ac.kr*
*Department of AI Convergence*
*Gwangju Institute of Science and Technology*

**Sejin Kim**                                           *sejinkim@gist.ac.kr*
*Department of AI Convergence*
*Gwangju Institute of Science and Technology*

**Sundong Kim**                                         *sundong@gist.ac.kr*
*Department of AI Convergence*
*Gwangju Institute of Science and Technology*

**Reviewed on OpenReview:** *https://openreview.net/forum?id=ULCOhBgGzy*

## Abstract

One of the core challenges in building general reasoning systems lies in generating diverse, human-aligned solution trajectories—different yet valid paths by which a problem can be solved. Prior approaches often rely on handcrafted templates, rule-based augmentations, or human demonstrations, which are limited in scalability and stylistic diversity. To address this, we explore the use of Generative Flow Networks (GFlowNets) for automated solution augmentation in reasoning tasks. We propose a framework that learns to generate diverse reasoning trajectories with probabilities proportional to their quality, guided by a human-inspired reward function and a novel geometric forward policy. This enables the generation of multiple plausible solution paths without relying on manual supervision. Moreover, our method supports efficient test-time augmentation from input-output examples alone, without access to ground-truth programs or external demonstrations—making it suitable for zero-shot settings. We evaluate our framework on the Abstraction and Reasoning Corpus (ARC-AGI), a benchmark designed to test compositional and abstract reasoning. Our results show that GFlowNets can effectively explore the space of valid reasoning processes, producing a variety of plausible reasoning trajectories, similar to how different individuals might solve the same problem using different intermediate steps. These trajectories are generated at scale—over 100k per task in under an hour, and follow a logarithmic yield trend, enabling practical tradeoffs between augmentation volume and novelty. Furthermore, fine-tuning a large language model (LLaMA 3.1 Instruct 8B) on these synthetic trajectories leads to a 28.6% improvement in reasoning accuracy on ARC tasks, demonstrating the downstream utility of our method. These findings suggest that GFlowNets offer a promising foundation for modeling structured reasoning in automated trajectory generation. Our code is here: `https://github.com/GIST-DSLab/GFN_to_ARC`

## 1 Introduction

One of the central challenges in developing intelligent systems is endowing them with the ability to reason—to solve novel problems by composing abstract concepts, drawing inferences, and planning intermediate steps.

Unlike perception or pattern recognition, which can often be improved through large-scale data and end-to-end optimization, reasoning requires an explicit modeling of multi-step, structured processes. In humans, reasoning is not just about arriving at correct conclusions, but about navigating diverse reasoning paths (trajectories) toward a solution. This diversity—different strategies, representations, or step sequences—plays a critical role in building a robust understanding and generalization. Hence, to improve reasoning capabilities in AI models, it is essential not only to optimize for the final answer, but also to model and expose the system to the wide range of processes by which those answers can be reached (Wei et al., 2022; Wang et al., 2023).

Recent research increasingly emphasizes the value of exposing language models to explicit reasoning trajectories—structured sequences of intermediate steps guiding problem-solving. This includes prompt-based strategies such as Chain-of-Thought (CoT) (Wei et al., 2022) and Tree-of-Thought (ToT) (Yao et al., 2023), sampling-based methods like Self-Consistency (Wang et al., 2023) and Best-of-N selection (Liu et al., 2025), and training-time approaches that filter or reward high-quality reasoning paths (Zelikman et al., 2022; Luo et al., 2024a; Wang & Su, 2015; Jiang et al., 2024). More recently, test-time iterative refinement has gained attention: models like Reflexion (Shinn et al., 2023), DeepSeek-R1 (DeepSeek-AI et al., 2025), and OpenAI's o1 (OpenAI, 2025; 2024) generate, evaluate, and revise their own reasoning traces in real-time. While these trends have yielded significant progress, most approaches still rely on prompt engineering, heuristic sampling, or post-hoc selection—limiting scalability and process-level alignment. In contrast, our work introduces a generative framework that learns to produce diverse reasoning trajectories in a unified and scalable manner. Our framework is unique in its ability to augment multiple, varied solution paths for a single problem instance, a capability lacking in existing human-collected or rule-based datasets (see Appendix D in detail.)

Existing methods for generating reasoning trajectories largely rely on two types of supervision: (1) human-written demonstrations and (2) rule-based augmentations. Human demonstrations (Strandgaard, 2024; LeGris et al., 2024; Kim et al., 2025b), often collected from crowdsourcing, capture authentic and diverse reasoning patterns. However, these solutions are costly to obtain, difficult to scale, and often noisy or inconsistent. More importantly, they are tied to specific training instances and cannot be applied to novel test tasks where such demonstrations are unavailable. In contrast, rule-based or hardcoded trajectories (Kim et al., 2024) offer expert-designed strategies that are typically consistent and efficient. These approaches rely on domain knowledge or templates to construct solution paths, making them effective within known task distributions. Yet their rigidity is also a limitation: they encode only a narrow slice of the solution space and fail to adapt to structurally novel problems. Furthermore, because they require access to ground-truth programs or transformation logic, they are inapplicable to unseen tasks at test time.

In many real-world reasoning scenarios, such as the ARC Prize (Chollet et al., 2025), systems must solve entirely unseen tasks at test time within a limited time budget (e.g., 6 minutes per task, assuming 120 tasks in 12 hours). Under these constraints, there is no opportunity to access task-specific supervision or hand-written demonstrations. This setting calls for test-time augmentation methods that can autonomously generate high-quality and diverse reasoning paths from scratch.

This leads us to a central research question: *Can we develop an automated framework that can generate diverse reasoning trajectories—without relying on human-written demonstrations or handcrafted rules—and use them to enhance model reasoning performance?* Such a framework would ideally be flexible enough to explore multiple plausible paths for the same problem, allowing the model to learn how to reason in more general, compositional, and interpretable ways. Moreover, it should be computationally efficient, able to generate thousands of diverse solution candidates within a short time window, enabling downstream symbolic or neural solvers to learn or select from these paths on-the-fly.

To evaluate the effectiveness of our proposed framework, we conduct experiments on the **Abstraction and Reasoning Corpus (ARC-AGI, hereafter ARC)** (Chollet, 2019), a benchmark specifically designed to test reasoning capabilities. ARC consists of few-shot, grid-based tasks where each instance requires the solver to infer an underlying transformation rule from a small number of input-output demonstrations. While the task format is visually simple, solving ARC problems often demands *compositional reasoning, abstract generalization, and logical coherence.* Indeed, ARC provides a unique lens through which to assess these dimensions of reasoning—*compositionality, productivity, logical coherence* (Lee et al., 2025), and the ability to generalize from limited examples.

Importantly, ARC's structure presents a particularly suitable environment for evaluating solution generation frameworks that aim to capture varied and structured reasoning paths, rather than simulating specific human-like strategies. First, the dataset is inherently underspecified: multiple reasoning trajectories can lead to the same correct output, making it ideal for exploring the benefits of diverse solution generation. Second, the limited data regime and absence of predefined task categories emphasize the need for flexible and generalizable reasoning strategies—exactly the kind our method aims to generate.

From an AGI perspective, this diversity in reasoning paths is not a peripheral detail but a central objective. Recent studies (Bengio, 2021; Morris et al., 2024; Kim & Kim, 2024; Lee et al., 2025) emphasize that general intelligence entails solving novel problems through varied, compositional reasoning processes, even when the final solution is unique. GFlowNets provide a principled framework for capturing such reasoning diversity. Instead of optimizing for a single expert path, they discover multiple plausible trajectories that lead to the same outcome. This capability aligns closely with the AGI goal of modeling not just what to think, but how to think through problems in multiple valid ways.

Finally, since each ARC task typically requires multi-step abstract inference rather than surface-level pattern recognition, improvements in trajectory-level modeling are more likely to translate into tangible performance gains. Our approach is also practically scalable: generating over 100k trajectories per task takes under an hour on average, and we observe a logarithmic growth trend in the discovery of new unique trajectories, allowing efficient tradeoffs between data volume and diversity. These properties make our method well-suited for test-time deployment in ARC-like reasoning scenarios.

**Contributions.** Our main contributions are summarized as follows:

1. **GFlowNet-Based Reasoning Augmentation:** We propose the first framework to use GFlowNets for generating diverse reasoning trajectories, enabling automated solution augmentation that reflects the structured variability in how problems can be solved.

2. **Human-Inspired Policy and Reward Design:** We introduce two architectural innovations—a geometric-forward action policy and a goal-conditioned reward function—that explicitly encode human reasoning biases into the learning process.

3. **Empirical Evaluation on ARC:** We demonstrate through extensive experiments on ARC that our method improves both solution diversity and accuracy compared to existing approaches, particularly in low-data regimes.

4. **Test-Time Efficiency and Yield Analysis:** We analyze the generation efficiency of GFlowNet and show that the number of unique, valid trajectories exhibits a logarithmic growth pattern, providing practical guidance on how much augmentation is beneficial under compute constraints.

## 2 Background

### 2.1 Reasoning with Diverse Solutions

Effective reasoning involves not just arriving at the correct answer, but also navigating structured and coherent intermediate steps. Many tasks—particularly those involving abstraction, logic, or multi-hop inference—admit multiple valid solution trajectories, each representing a different way to solve the same problem. This diversity is analogous to the variation in strategies employed by humans and is essential for improving generalization, robustness, and interpretability in AI systems.

To incorporate this inductive bias, recent research has introduced techniques that explicitly model the reasoning process. Chain-of-Thought (CoT) prompting (Wei et al., 2022; Zhang et al., 2025a), Tree-of-Thought (ToT) (Yao et al., 2023), and Process Reward Models (PRMs) (Zhang et al., 2025b) supervise or sample intermediate reasoning steps to better align with human-like cognition. These methods show that exposing models to diverse reasoning paths improves their ability to generalize, particularly in tasks with sparse supervision or structural novelty. Additional evidence across various reasoning benchmarks—including GSM8K (Cobbe et al., 2021), SVAMP, and AQuA-RAT (Ling et al., 2017)—further supports this approach,

demonstrating consistent performance improvements when models explore and aggregate multiple reasoning trajectories.

Complementary approaches such as automated process supervision (Luo et al., 2024a) and planning-based trajectory optimization (Jiang et al., 2024) further validate the effectiveness of learning from reasoning paths directly. These methods synthesize reward signals that capture desirable reasoning characteristics–such as conciseness, non-redundancy, or clear subgoal structure–without relying on exhaustive human annotations. Similarly, reasoning-driven process reward modeling (She et al., 2025), and step-by-step human feedback (Lightman et al., 2024) have achieved substantial performance gains by leveraging automated or semi-automated trajectory supervision. As a result, models trained with these signals exhibit more structured and interpretable problem-solving behavior, underscoring a growing consensus that trajectory-level supervision significantly enhances reasoning abilities.

This emphasis on process-level reasoning has also become a central trend in the development of large language models (LLMs). Recent models like OpenAI's *o3* series (OpenAI, 2025) and DeepSeek's *R1* (DeepSeek-AI et al., 2025) incorporate internal deliberation and iterative refinement to improve reasoning performance. Likewise, Self-Consistency sampling (Wang et al., 2023) aggregates multiple reasoning chains through majority voting, achieving improved accuracy and robustness on benchmarks such as GSM8K (Cobbe et al., 2021). These examples highlight a growing consensus: modeling diverse and structured reasoning processes is key to achieving reliable and human-aligned inference.

However, many of these approaches depend on human-written demonstrations, handcrafted rules, or prompt-based sampling, which constrain scalability and stylistic diversity. Moreover, outcome-only supervision fails to distinguish between logically correct but inefficient reasoning paths and concise, interpretable ones. Without process-level reward signals, models are unable to prioritize solution trajectories that mirror human preferences.

These limitations motivate our goal of building a generative framework that can learn to produce diverse, high-quality reasoning trajectories in a scalable and automated manner. In this work, we leverage Generative Flow Networks (GFlowNets) (Bengio et al., 2021; 2023) to model solution generation as a reward-proportional sampling problem, enabling the creation of multiple human-aligned reasoning paths without requiring explicit supervision for each trajectory.

### 2.2 Data Augmentation for Reasoning Tasks

Data augmentation addresses the inherent scarcity of supervision in reasoning tasks and can be broadly categorized into input-output augmentation and solution-process (trajectory) augmentation.

**Input-Output Pair Augmentation.** Methods such as RE-ARC (Hodel, 2024), AugARC (Bikov et al., 2025), and SOLAR (Kim et al., 2024) generate new input-output examples to enhance training diversity in ARC tasks. Similar strategies, including synthetic data generation and paraphrasing, have been widely applied across various reasoning domains. For math word problems, Lu et al. (2024) leveraged question back-translation to produce synthetic examples, significantly enhancing model performance on the GSM8K benchmark. In logical reasoning, LogicAsker generated QA pairs by programmatically translating formal logic into natural language, achieving notable gains in logical consistency (Wan et al., 2024). Likewise, in code generation tasks, WizardCoder synthesized diverse coding problems and solutions, leading to state-of-the-art results on benchmarks like HumanEval and MBPP (Luo et al., 2024b). For scientific reasoning tasks, cross-lingual data augmentation using large language models produced multilingual synthetic QA pairs, demonstrating considerable improvements in cross-lingual generalization (Whitehouse et al., 2023). These input-output augmentation techniques substantially improve model generalization by diversifying the task distribution, though they typically do not explicitly expose underlying reasoning processes.

**Solution-Process (Trajectory) Augmentation.** Techniques explicitly augmenting reasoning trajectories leverage models' intermediate solution paths. Self-Consistency sampling (Wang et al., 2023), Tree-of-Thought deliberative search (Yao et al., 2023), and iterative refinement methods like Reflexion (Shinn et al., 2023) exemplify trajectory-level augmentation. Methods such as STaR (Zelikman et al., 2022) and Automate-

CoT (Shum et al., 2023) demonstrated that training on model-generated chains-of-thought significantly boosted accuracy across various benchmarks, including commonsense and math word problems. Synthetic Prompting (Shao et al., 2023) further enriched trajectory diversity by iteratively synthesizing questions and corresponding reasoning paths, yielding substantial gains in numerical and algorithmic reasoning tasks. Similarly, frameworks like Flow of Reasoning (Yu et al., 2024) leveraged planning-based trajectory sampling to explore diverse reasoning paths, surpassing previous state-of-the-art performances on puzzle-solving benchmarks. AlphaCode's strategy of sampling numerous candidate solutions and evaluating correctness tests illustrates another successful trajectory-level augmentation approach in code generation tasks (Li et al., 2022). Collectively, these trajectory-focused methods enable models to explore and validate multiple solution paths systematically, greatly enhancing performance and interpretability.

**Other Augmentation Techniques.** Best-of-N (BoN) sampling and ensemble decoding represent additional augmentation strategies. BoN methods, applied in models like Minerva and AlphaCode, sample multiple candidate outputs to select the best-performing solution (Lewkowycz et al., 2022; Li et al., 2022). Ensemble decoding aggregates outputs from multiple models or reasoning chains, typically employing voting or averaging strategies. These methods primarily function at the output level and rely heavily on external evaluation metrics or heuristic criteria.

**Limitations and Challenges.** Despite their effectiveness, existing augmentation methods face challenges including computational scalability, genuine trajectory diversity, and the absence of reliable intermediate-step supervision. Techniques such as Tree-of-Thought and extensive sampling-based methods incur significant computational overhead (Yao et al., 2023; Wang et al., 2023). Furthermore, approaches relying on outcome-only evaluation lack explicit guidance for intermediate reasoning steps, potentially causing logical inconsistencies or inefficient reasoning pathways.

Overall, integrating diverse augmentation strategies—ranging from input-output pair expansion to trajectory-level solution enhancement—can significantly improve model robustness and generalizability. These methods collectively form an essential toolkit for addressing the inherent limitations of scarce supervision in complex reasoning tasks like ARC and beyond.

## 2.3 Generative Flow Networks (GFlowNets)

Generative Flow Networks (GFlowNets) (Bengio et al., 2021; 2023) are a class of generative models designed to produce diverse solution trajectories by learning from reward signals. They are particularly effective for structured generation problems where multiple valid outputs exist, such as generating human-like solutions in reasoning tasks.

Unlike standard generative models, which often produce one-shot outputs, GFlowNets construct solutions incrementally by sampling sequences of actions. This approach is inspired by reinforcement learning (RL), enabling GFlowNets to combine the exploration strengths of RL with the flexibility of generative modeling.

In contrast to conventional RL, which typically seeks a single optimal policy, GFlowNets learn a stochastic policy that assigns higher probability to trajectories with higher terminal rewards. This makes them well-suited for sparse-reward environments and problems with multiple equally good solutions, such as ARC. GFlowNets have demonstrated their utility across a wide range of structured generation tasks, including molecular design and biosequence modeling (Bengio et al., 2021; Gaiński et al., 2025; Jain et al., 2022; Jang et al., 2024; Zhang et al., 2022), symbolic regression (Zhang et al., 2023b), combinatorial optimization (Jain et al., 2023; Zhang et al., 2023a), and other applications (Sendera et al., 2024; Zhang et al., 2022; Pan et al., 2023; Falet et al., 2024; Seo et al., 2025; Kim et al., 2025a; Hu et al., 2024).

**Flow Matching Condition and Trajectory Balance (TB) Loss.** To understand the underlying mechanism of GFlowNets, it is essential to grasp the *flow matching condition*, which maintains consistency between the inflow and outflow of probability mass at any state in the state space:

$$\sum_{s'} F(s' \to s) = \sum_{s''} F(s \to s''), \tag{1}$$

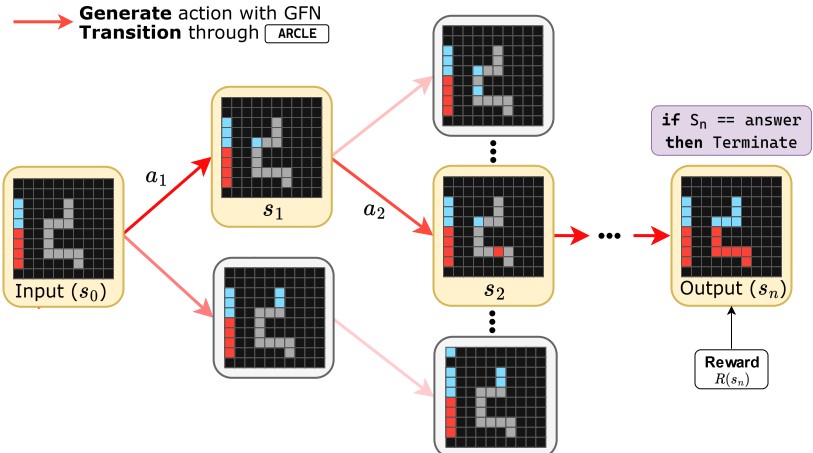

Figure 1: GFlowNet Concept Diagram for Solution Generation in ARC

where $F(s' \rightarrow s)$ denotes the probability flow from state $s'$ to state $s$. This condition ensures that the forward policy $P_F(s_t \mid s_{t-1})$ (governing state transitions from an initial state to terminal states) and the backward policy $P_B(s_{t-1} \mid s_t)$ (governing transitions in reverse) remain consistent with the reward model's distribution.

This consistency is operationalized through the *Trajectory Balance (TB) loss* (Malkin et al., 2022), defined as:

$$Z \prod_{t=1}^{n} P_F(s_t \mid s_{t-1}) \; = \; R(x) \prod_{t=1}^{n} P_B(s_{t-1} \mid s_t), \tag{2}$$

where $R(x)$ represents the reward for a trajectory $x = (s_0, \ldots, s_n)$, and $Z$ is a learnable normalization constant ensuring proper distribution over trajectories. Minimizing the squared difference of the log probabilities on both sides yields the TB loss:

$$\mathcal{L}_{\text{TB}}(\theta) = \left( \log Z_\theta + \sum_{t=1}^{n} \log P_F(s_t \mid s_{t-1}; \theta) - \log R(x) - \sum_{t=1}^{n} \log P_B(s_{t-1} \mid s_t; \theta) \right)^2. \tag{3}$$

By optimizing this objective, GFlowNets learn to sample trajectories proportionally to their rewards, naturally prioritizing high-quality trajectories. This allows effective exploration of complex solution spaces.

**Training Steps and DAG Structure.** The practical training loop of GFlowNets involves several key steps: (1) *Forward Sampling*, starting from an initial state and sequentially sampling states via $P_F$ until a terminal state is reached; (2) *Reward Computation*, evaluating the sampled trajectory using the reward function $R(x)$; (3) *Backward Sampling*, optionally reconstructing the trajectory in reverse via $P_B$ to reinforce flow matching; and (4) *Loss Computation and Parameter Update*, adjusting model parameters by minimizing the TB loss. To ensure stable training and avoid cyclic or infinite loops, state transitions are typically constrained within a Directed Acyclic Graph (DAG), limiting each state to a single visitation and ensuring convergence.

**Reward Modeling and Oracle Networks.** Since GFlowNets explicitly learn the distribution shaped by the reward function, accurate reward modeling is paramount. Often, rewards are assigned through pre-trained or heuristic oracle networks tailored to specific tasks. These oracle models encapsulate task-specific structural insights and desired properties, significantly enhancing the effectiveness of the generative process. For instance, in molecular generation tasks, oracle networks evaluate chemical viability, steering GFlowNet exploration toward feasible and high-quality solutions. The careful design and training of these oracle networks thus play a critical role in the performance of GFlowNets, particularly in environments with sparse or complex reward structures.

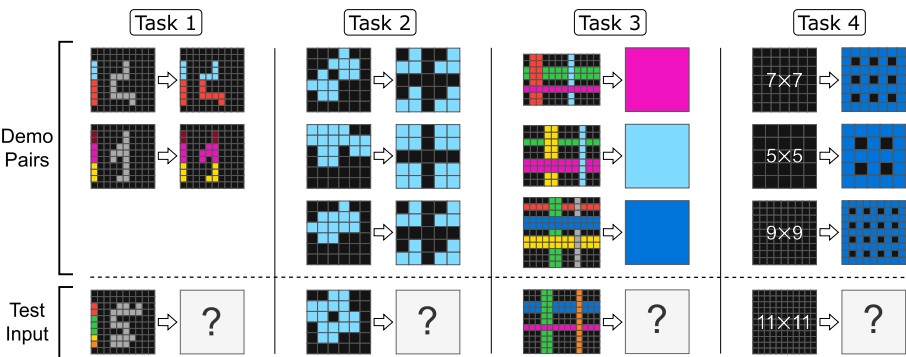

Figure 2: Some Problems of Abstraction and Reasoning Corpus (ARC)

The structural exploration capability and explicit reward modeling of GFlowNets position them uniquely to capture intricate reasoning structures and produce diverse, valid solutions in reasoning tasks. By leveraging human-inspired reward models, GFlowNets can systematically generate solution trajectories reflective of human reasoning patterns. This capability makes GFlowNets particularly suited for automated trajectory augmentation tasks, such as those required by reasoning-intensive benchmarks like ARC.

Specifically, GFlowNets directly support logical coherence through forward and backward flow consistency, ensuring generated solutions maintain logical consistency. Productivity is enabled by effective generalization from sparse examples via oracle guidance, and compositionality is facilitated by structurally exploring compositional reasoning paths evaluated by oracles.

## 2.4 ARC as a Benchmark for Process-Level Reasoning

The Abstraction and Reasoning Corpus (ARC) (Chollet, 2019) is a challenging benchmark comprising visual grid-based puzzles that require inferring abstract and compositional rules from very few provided examples. Each ARC task includes only 2–5 input-output examples, with no repetition of input-output pairs across different tasks. The ARC dataset was originally introduced as a proxy for evaluating core reasoning abilities of AI systems, specifically in terms of logical coherence, productivity, and compositionality (Lee et al., 2025).

Recently, despite impressive performances by state-of-the-art models such as OpenAI's O3–low, which achieved over 75% accuracy on ARC (OpenAI, 2025), performance sharply declined to below 10% on the newly introduced ARC-AGI-2 dataset (Chollet, 2024; Kamradt, 2025). This drop clearly indicates the complexity of ARC and highlights the significant gap between current AI capabilities and human-like abstraction and reasoning, indirectly underscoring the distance to achieving Artificial General Intelligence (AGI).

The inherent difficulty of ARC arises primarily from its unique structure: each problem set has a single, unique input-output relationship, and these relationships vary drastically across different tasks. Therefore, conventional data augmentation methods struggle as additional synthetic examples cannot be trivially generated. Given these constraints, ARC becomes particularly suitable for testing the capability of automated solution augmentation methods. Specifically, ARC provides a rigorous environment to examine whether generative frameworks, such as GFlowNets, can effectively capture structural insights from carefully designed, human-inspired reward models, thus facilitating the discovery of diverse and valid solution trajectories under highly sparse and unique example constraints. These characteristics position ARC as an ideal benchmark for exploring and validating advanced reasoning frameworks like GFlowNets.

Conventional augmentation methods applied specifically to ARC, such as input-output pair augmentation ((Hodel, 2024; Bikov et al., 2025; Kim et al., 2024)), lack explicit trajectory-level reasoning, treating the transformation processes as black boxes. Rule-based augmentation methods, meanwhile, rely on predefined human-crafted rules, severely limiting scalability and the generalization capacity to novel, unseen problem structures. Consequently, these approaches fail to adequately address ARC's unique demands for explicit and

diverse trajectory generation, highlighting the need for automated, trajectory-level augmentation frameworks such as GFlowNets.

# 3 Methods

## 3.1 GFlowNet Architecture for Human-Aligned Solution Generation in ARC

We now provide a concise overview of how our GFlowNet is structured and generates solutions tailored specifically for ARC problems. Building on the general components described in Sections 3.1–3.2, our architecture differs from conventional GFlowNet models primarily by using asymmetric sampling distributions: a geometric distribution for forward sampling to prioritize shorter, more efficient action sequences, and a categorical distribution for backward sampling to ensure effective trajectory reconstruction. Additionally, we explicitly incorporate human trajectory priors into our reward function, penalizing redundant or excessively long trajectories to enhance learning efficiency and promote human-like reasoning. In what follows, we detail this architecture and elaborate on the rationale behind these adaptations.

**High-Level Process.**  At the start of each episode, an initial state derived from the ARC input grid is set, containing minimal information needed for the ARC task. The GFlowNet forward policy then samples actions to move from one grid configuration to another, guided by the *Geometric* distribution described above. Actions that prove too long or repetitive are penalized by our reward model, while succinct, high-quality trajectories receive higher rewards. After reaching a final state, the backward policy verifies or backtracks using a *Categorical* distribution, helping to ensure the global flow matching necessary for Trajectory Balance. Over many episodes, this bidirectional sampling process adapts the GFlowNet parameters to favor human-like solutions that meet the puzzle's requirements. We summarize the full GFlowNet training process in Algorithm 1, provided in Appendix C. The pseudocode highlights the use of geometric sampling for forward actions, categorical backward modeling, and off-policy updates with cycle-penalized reward shaping.

**Algorithmic Flow.**  Algorithm 1 (Appendix C) outlines the main steps: *(1) Initialization* of the forward/backward networks, *(2) Forward Sampling* where actions are chosen via the Geometric distribution based on predicted logits, *(3) Reward Assignment* using a human-inspired reward model that applies discounting, cycle penalties, trajectory-length constraints, or a combination thereof, depending on the specific characteristics of the trajectory, *(4) Backward Sampling* where the backward policy samples a probable reverse action corresponding to the forward-selected action, ensuring local consistency required for trajectory balance, and *(5) Parameter Update* by minimizing the trajectory balance (TB) loss across the sampled trajectory. Repeating these steps allows the GFlowNet to efficiently converge on a diverse set of high-reward, human-like trajectories.

**ARC-Specific Adaptations.**  Unlike many domains where actions naturally exhibit uniform or categorical distributions, ARC problems benefit from the combination of a *Geometric-forward* and *Categorical-backward* design. (Section 3.2). In our setup, each trajectory typically ends with a "Submit" action that finalizes the generated grid, triggering a binary reward if the puzzle is solved correctly. This setup not only aligns with the binary reward structure of ARC but also enables GFlowNet to efficiently prioritize high-quality, human-like trajectories within its generative process.

**Test-Time Augmentability.**  A key advantage of our framework is its ability to generate solution trajectories directly from input-output demonstrations, without requiring access to ground-truth programs or human-written paths. This enables reasoning augmentation even at test time, where neither human demonstrations nor rule-based solutions are available. In contrast to prior methods that rely on fixed trajectory sources tied to specific training tasks, our approach can synthesize plausible and diverse reasoning paths on unseen problems. **This is particularly important in settings like ARC, where the model must solve novel tasks at test time without access to prior task-specific supervision.** As a result, our method is especially well-suited for ARC's few-shot, out-of-distribution generalization setting. For detailed empirical analysis of efficiency and test-time augmentability (e.g., trajectory generation rate and yield scaling), please refer to Appendix E.

### 3.2 Human-Trajectory Guided Reward Design

GFlowNet cannot effectively learn with a purely sparse reward model. A naive sparse reward assignment—1 for correct answers and 0 for incorrect ones—fails to provide the necessary gradient to differentiate between high-quality and low-quality trajectories, severely limiting learning efficiency or even halting progress entirely.

When applying GFlowNet to the ARC problem, this limitation became apparent. For example, as shown in Figure 3, some trajectories solve the problem efficiently, while others reach the correct answer with unnecessary and repetitive actions. Assigning the same reward to both types of trajectories makes it difficult for GFlowNet to prioritize high-quality solutions.

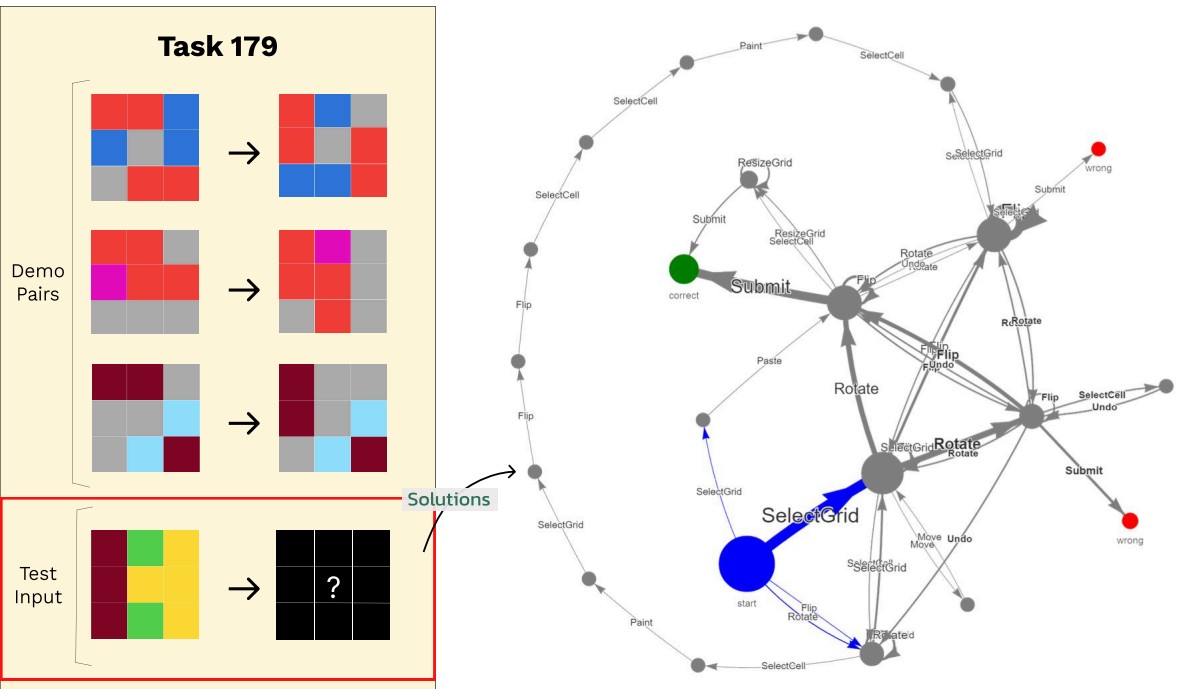

Figure 3: **State Space Graph of Trajectories for ARC Problem**. The trajectories were collected through O2ARC (Shim et al., 2024), illustrating various solution paths from the start node (blue) to the correct solution node (green). Additionally, incorrect solution nodes (red) are displayed to represent alternative, unsuccessful attempts. This figure highlights the diversity of possible solutions for a single problem, emphasizing the multiple ways in which the problem can be approached and solved.

To address this, we analyzed human trajectory data from O2ARC 3.0 (Shim et al., 2024) to identify clear quality differences and inform the design of a reward model that highlights superior trajectories. Two major characteristics of high-quality trajectories were identified:

- **Final Action:** The last action in a trajectory must always be the '*Submit*' action. This deliberate conclusion indicates that the solution is complete and allows for feedback. Trajectories that end with the Submit action are prioritized in the reward model.

- **Avoidance of Redundancy:** High-quality trajectories avoid unnecessary repetition, using only essential actions to reach the correct answer. Most general problems rarely require more than 10 steps (Kim et al., 2025b). Repetitive actions, particularly in a Directed Acyclic Graph (DAG) context, can create cycles that complicate learning.

To reflect these human-derived structural preferences explicitly, we propose integrating the following mechanisms into our reward design:

**Discount Factor ($R_{\textbf{DF}}$)**  The first approach involves applying a discount factor $\gamma$ (set to 0.9), commonly used in reinforcement learning. This method inherently favors shorter trajectories by scaling the reward $R$ at each step $t$ as:

$$R_{\mathrm{DF}} = R \cdot \gamma^t \tag{4}$$

While this mechanism promotes brevity, our goal is not strict minimization, as our experiments show this can prematurely limit the discovery of diverse solutions. Rather, this approach primarily serves as a heuristic to guide the search away from aimless wandering and toward more direct, goal-oriented trajectories.

**Trajectory Length Regularization ($\mathcal{L}_{\textbf{TLR}}$)**  The second strategy introduces a regularization term directly into the objective function. Offering a more flexible way to manage exploration depth, this term penalizes significant deviations from a target trajectory length $\mathcal{L}_{\mathrm{target}}$. This provides a soft constraint that discourages excessively long or inefficient paths without being overly restrictive, providing flexibility while discouraging excessively long trajectories:

$$\mathcal{L}_{\mathrm{TLR}} = \lambda_{\mathrm{reg}} \cdot (\mathcal{L}_\tau - \mathcal{L}_{\mathrm{target}})^2 \tag{5}$$

Here, $\lambda_{\mathrm{reg}}$ controls the regularization strength. The total objective function combines this term with the Trajectory Balance Loss ($\mathcal{L}_{\mathrm{TB}}$):

$$\mathcal{L}_{Total}(\theta) = (1 - \alpha) \cdot \mathcal{L}_{\mathrm{TB}} + \alpha \cdot \mathcal{L}_{\mathrm{TLR}} \tag{6}$$

where $\alpha$ balances trajectory balance and length regularization.

**Cycle Detection ($R_{\textbf{cycle}}$)**  Finally, cycle detection penalizes repetitive actions by flagging repeated states within a trajectory. A penalty proportional to the number of detected cycles $C(\tau)$ is applied:

$$R_{cycle}(\tau) = \begin{cases} r(S_T) - \lambda \cdot C(\tau) & \text{if a cycle is detected,} \\ r(S_T) & \text{otherwise.} \end{cases} \tag{7}$$

This encourages efficient exploration while avoiding redundant actions.

Thus, we define the reward for a trajectory $x = (s_0, \ldots, s_T)$ as:

$$R(x; y) = \mathbf{1}_{[f(x)=y]} \cdot \gamma^{L(x)} \cdot \exp\left(-\lambda_{\mathrm{cycle}} \cdot C(x)\right),$$

where:

- $\mathbf{1}_{[f(x)=y]}$ is a binary indicator of whether the trajectory leads to a correct solution,
- $\gamma \in (0, 1)$ is a discount factor applied to the trajectory length $L(x)$,
- $C(x)$ is the number of detected cycles (i.e., repeated states) in the trajectory,
- $\lambda_{\mathrm{cycle}}$ is the penalty weight for cyclic behavior.

This reward is used as input to the Trajectory Balance (TB) loss, and is log-transformed prior to optimization.

In addition to the reward-based objective, we introduce a separate trajectory length regularization loss to further encourage concise reasoning:

$$\mathcal{L}_{\mathrm{TLR}}(x) = \lambda_{\mathrm{TLR}} \cdot (L(x) - \mathcal{L}_{\mathrm{target}})^2,$$

where $\mathcal{L}_{\mathrm{target}}$ is a reference trajectory length (e.g., 3 or 4), and $\lambda_{\mathrm{TLR}}$ controls the penalty strength.

The overall training objective is then defined as:

$$\mathcal{L}_{\text{total}} = \mathcal{L}_{\text{TB}} + \mathcal{L}_{\text{TLR}},$$

where $\mathcal{L}_{\text{TB}}$ denotes the Trajectory Balance loss computed using the log-transformed reward $\log R(x; y)$.

### 3.3 Forward Policy via Geometric Action Sampling

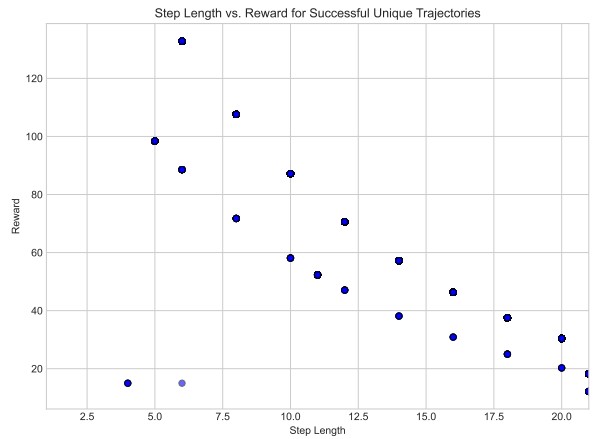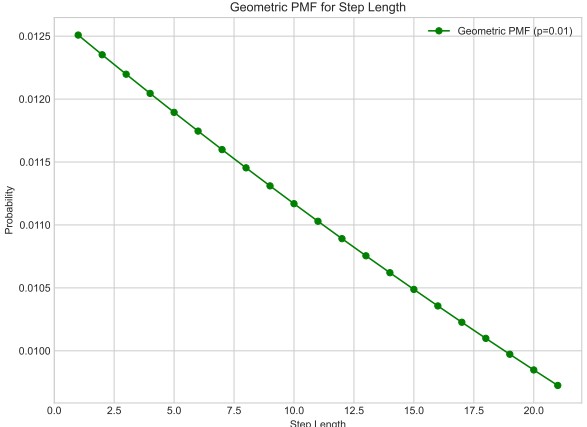

Figure 4: (Left) **Reward Distribution by Step Length**: This figure shows the reward distribution across trajectories with varying lengths for ARC problems. Higher rewards are concentrated in shorter steps, while rewards decrease as the step length increases. (Right) **Geometric PMF**: General form of PMF for the Geometric distribution.

To effectively model the distribution of reward functions derived from human trajectories, it is crucial to account for the sequential nature of actions and their dependencies in ARC problems. While conventional GFlowNets typically employ the Categorical distribution to model action probabilities, this approach fails to capture the observed relationship between reward and step length in ARC problems.

As shown in Figure 4, rewards are heavily skewed toward shorter trajectories, with diminishing rewards as step lengths increase. This motivates the use of the Geometric distribution to address the limitations of the Categorical model.

The Geometric distribution is well-suited for modeling action sequences under a framework where actions are treated as probabilistic steps toward success. Specifically, it represents the number of steps required until the first success in a sequence of trials, with a constant probability of success $p$. In the context of ARC problems, success can be interpreted as reaching the target state, and each action step is viewed as a probabilistic attempt to move closer to that state.

To formalize this interpretation, we treat each candidate action $a_i$ as associated with a success probability $p_i$ obtained via softmax over logits $z_i$. The number of steps until success under this distribution is modeled as a geometric random variable:

$$X_i \sim \text{Geometric}(p_i), \quad \mathbb{E}[X_i] = \frac{1}{p_i}.$$

Then, selecting the action with the minimum expected steps corresponds to:

$$a^* = \arg\min_i \mathbb{E}[X_i] = \arg\max_i p_i.$$

This provides a probabilistic justification for the use of Geometric sampling in the forward policy: it naturally prioritizes actions with higher expected efficiency in sparse reward settings. Moreover, by selecting actions based on their expected time-to-success, the policy aligns with human-like decision-making that favors shorter and more effective trajectories.

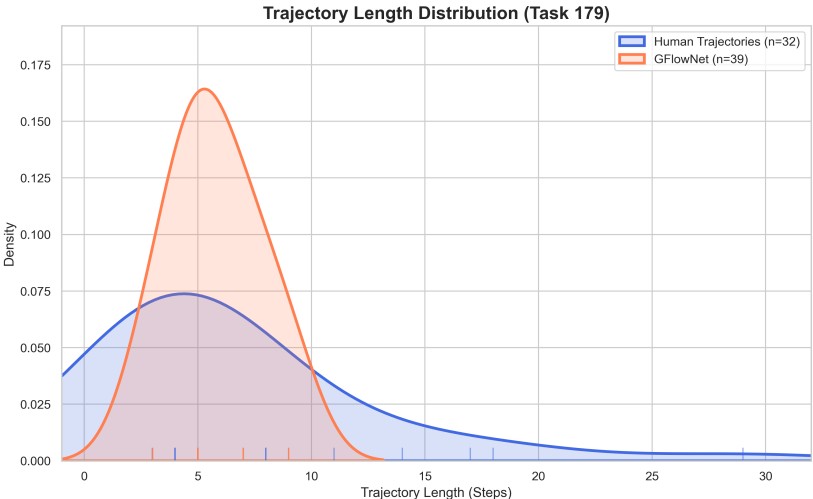

Figure 5: Comparison of correct and unique trajectory length distributions between human-provided solutions (n=32) and our GFlowNet-generated solutions (n=39) for Task 179 (ID: 74dd1130). The GFlowNet's distribution (orange), guided by the geometric policy, shows a strong preference for shorter trajectories, peaking at a similar length to the human distribution (blue), empirically validating our brevity-focused heuristic.

This aligns well with the structure of GFlowNet, where the model explores action sequences that maximize long-term reward under trajectory-level uncertainty. Building on the efficiency bias introduced by geometric sampling, this formulation has proven particularly effective in sparse-reward environments such as ARC. In addition, the preference toward shorter trajectories helps mitigate over-exploration of redundant or excessively long paths, which can dilute learning signals especially during early stages of training.

This theoretical bias toward shorter, high-probability trajectories is grounded in established principles like Minimum Description Length (MDL) (Hernandez-Orallo & Minaya-Collado, 1998; Mahoney, 1999; Hutter, 2005; Legg & Hutter, 2007; of Ockham, 1990). This preference for conciseness acts as a powerful heuristic, and it aligns with general syntactic patterns observed in human problem-solving trajectories, as illustrated in Figure 5, our model learns to generate trajectories with a length distribution that closely mirrors that of human solutions, with both distributions peaking at a small number of actions. This provides strong empirical support for our design choice to incorporate an inductive bias towards brevity, validating that the model effectively captures key structural features of efficient, human-like reasoning paths.

However, it is crucial to recognize that this heuristic is a means to an end, not an absolute objective. As our detailed analysis on exploration depth will demonstrate in 4.2.2, treating brevity as the sole goal is a limiting assumption. The true strength of our policy emerges when its efficiency bias is carefully balanced with a sufficient 'exploration budget' of steps. This balance is key to preventing premature convergence and enabling the discovery of more diverse and complex reasoning paths.

The rationale for using the Geometric distribution is as follows:

- **Reward Trends in ARC problems**: Higher rewards are observed in shorter trajectories, which can be naturally modeled by the Geometric distribution's property of assigning higher probabilities to fewer steps. By prioritizing trajectories with shorter action sequences, the model effectively aligns exploration with the observed reward trends in human problem-solving.

- **Action Sequences as Trials**: Each action in a sequence can be interpreted as an independent attempt to reach the target state (success). The Geometric distribution allows the model to evaluate and prioritize actions based on their probabilities of success, guiding the GFlowNet to focus on trajectories that are both efficient and high-reward.

To integrate the Geometric distribution into GFlowNet's forward policy, the policy network outputs a logits vector $\mathbf{z} = (z_1, z_2, \ldots, z_N)$ for a given state $\mathbf{s}$. This vector is transformed into probabilities $\mathbf{p} = (p_1, p_2, \ldots, p_N)$ via the softmax function. For each action $a_i$, the probability of success is modeled as a Geometric random variable $X_i \sim \text{Geometric}(p_i)$, where the selection of an action $a^*$ is defined as:

$$a^* = \arg\min_i X_i. \tag{8}$$

Here, the action with the fewest expected steps to success is prioritized, effectively capturing the reward distribution's dependency on step length and enhancing the exploration process. This step-based probability assignment allows GFlowNet to balance between exploration and exploitation, focusing on high-reward trajectories while maintaining diversity.

Conversely, the backward policy retains the use of the Categorical distribution. While the Geometric distribution is advantageous for exploration, its properties are less suited for reconstructing paths from the goal to the start state. The backward policy instead relies on the Categorical distribution to guide path regression, where capturing dependencies between actions is less critical.

Attempts to apply the Geometric distribution to the backward policy yielded suboptimal results, as it failed to represent the necessary dependencies for effective path reconstruction. Therefore, the use of distinct distributions—Geometric for forward exploration and Categorical for backward reconstruction—is both empirically justified and theoretically grounded.

## 4 Experiments

### 4.1 Experimental Design

#### 4.1.1 Research Questions

Our central goal is to explore whether Generative Flow Networks (GFlowNets) can be adapted to generate reasoning trajectories that are **aligned with human problem-solving heuristics** in the ARC domain. Since direct comparison with human trajectories is beyond our current scope, we instead assess *alignment* through structural properties inspired by human behavior (e.g., brevity, intentionality, loop avoidance). To this end, we pose the following sub-questions:

- **RQ1. Can reward functions informed by human priors promote human-aligned reasoning patterns?** We assess how different reward shaping techniques (e.g., brevity encouragement, loop avoidance, trajectory regularization) influence the structure and quality of generated trajectories.

- **RQ2. Does using a Geometric distribution in the forward policy enhance efficiency than conventional GFlowNets with a categorical forward policy in diverse trajectory generation?** We evaluate whether modeling forward decisions via the Geometric distribution, which prioritizes concise action sequences, improves alignment with diverse reasoning behaviors.

- **RQ3. Why does a Geometric forward policy excel in goal-conditioned reasoning tasks?** We compare forward–backward policy combinations (G–C, G–G, C–G, C–C) to uncover how the inductive bias of a Geometric forward policy, together with a stabilizing backward policy, leads to higher success and diversity under sparse rewards.

### 4.1.2  Dataset: Task Selection Rationale

The dataset used in our experiments is based on the Abstraction and Reasoning Corpus (ARC), specifically focusing on Task 179 (*Diagonal Flip*) for most experiments. This task exemplifies problems where the solution requires transformation of the entire grid, rather than localized subregions, making it suitable for evaluating solution augmentation in full-grid reasoning contexts.

Formally, the selected tasks satisfy:

$$f(x_k) = y_k, \quad S = G, \quad \text{where } G \in \mathbb{R}^{m \times n} \tag{9}$$

Here, $S$ denotes the selection region and $G$ represents the entire grid with dimensions $m \times n$. By defining $S = G$, each action operates on the whole grid, requiring full-grid comprehension and transformation.

To further constrain the solution space and evaluate the impact of full-grid transformations, we restrict our experiments to ARC problems solvable via *entire-grid* operations—such as flips, rotations, translations, and global color replacements. This design choice is both practically and theoretically motivated. In the ARCLE environment (Lee et al., 2024), the default formulation allows any subset of the grid—including individual pixels or subgrids—to be transformed. However, this leads to an exponentially large action space of roughly $O(A^L \times 2^{mn})$ for a grid of size $m \times n$, where $A$ is the set of atomic actions and $L$ is the typical sequence length. For a $30 \times 30$ grid, the subregion selection space alone reaches $2^{900}$, rendering effective exploration intractable. By instead focusing on whole-grid transformations (i.e., $S = G$), we reduce the action space complexity to approximately $O(A^L)$, which not only enhances tractability but also aligns more closely with the structural reasoning patterns observed in human trajectories.

In addition to Task 179, other tasks solvable by whole-grid selection include: 53, 87, 129, 140, 150, 155, 241, and 380 are tested. These are shown in appendix B.3.

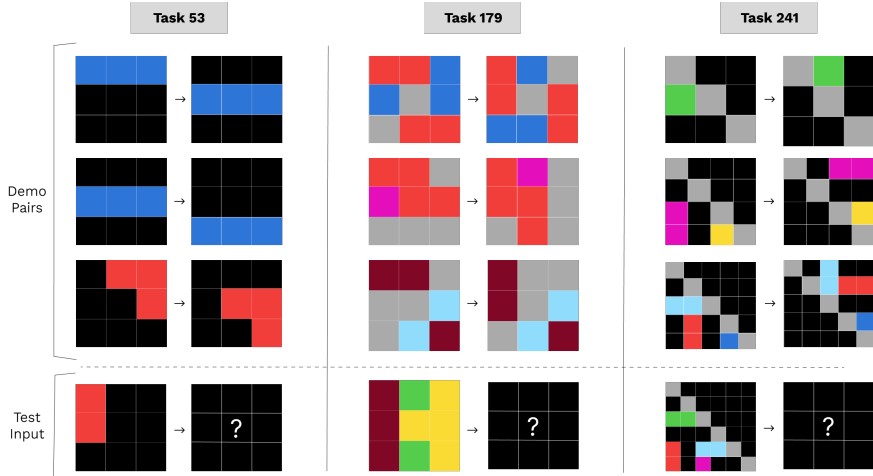

Figure 6: Example of an Entire grid selection ARC problem

### 4.1.3  Evaluation Metrics

The following metrics are used to evaluate the performance of GFlowNet:

- **Validation Accuracy (Val_ACC)**: The proportion of correctly generated trajectories among 100 attempts. A trajectory is considered correct if it reaches the target solution while adhering to the task constraints. This metric evaluates the model's ability to generate valid and successful trajectories efficiently.

- **Trajectory Diversity ($D_{\mathbf{traj}}$)**: The fraction of unique successful trajectories among all generated unique trajectories. A higher diversity score indicates that the model explores multiple valid solutions

rather than converging on a single mode. This reflects the model's ability to balance exploration and exploitation in trajectory generation.

- **Reward Distribution Diversity ($D_{\mathbf{reward}}$)**: The variation in assigned rewards, measured using the Shannon Index. A higher value suggests that the model assigns rewards more evenly across different trajectories, encouraging diverse exploration. Conversely, a lower value indicates that rewards are concentrated on a limited set of trajectories, leading to mode collapse.

### 4.1.4 Hyperparameters and Glossary

Below are Table 1, which presents key hyperparameter settings, and Table 2, which defines key abbreviations used throughout the paper.

| Hyperparameter | Value |
| --- | --- |
| Learning Rate (lr) | $10^{-4}$ |
| # of Actions | 5 |
| Episode Length | 10 |
| Base Reward | 15 (O), 0 (X) |
| Discount Factor ($\gamma$) | 0.9 |
| Loss Weight ($\alpha$) | 0.2 |
| Trajectory Regularization Weight | 0.01 |

Table 1: Key Hyperparameter Settings

| Abbreviation | Definition |
| --- | --- |
| $P_F$ | Forward Probability |
| $P_B$ | Backward Probability |
| TLR | Trajectory Length Regularization |
| DF | Discount Factor |
| cycle | Cycle Penalty Reward |
| Splus | Submit plus Reward |
| $\mathcal{L}_{\mathrm{TB}}$ | Trajectory Balance Loss |

Table 2: Glossary of Terms and Abbreviations

## 4.2 Performance of Revised GFlowNet for Solution Augmentation

This section evaluates the effectiveness of our proposed GFlowNet modifications across the three research questions defined in Section 4.1. To qualitatively illustrate the generated trajectories discussed throughout our experiments, we present examples from Task 150 (ID: 7a3c6ac) and Task 179 (ID: 74dd1130)—two ARC problems included in our benchmark suite (Appendix B.3). These examples highlight the variety of transformation sequences leading to correct solutions.

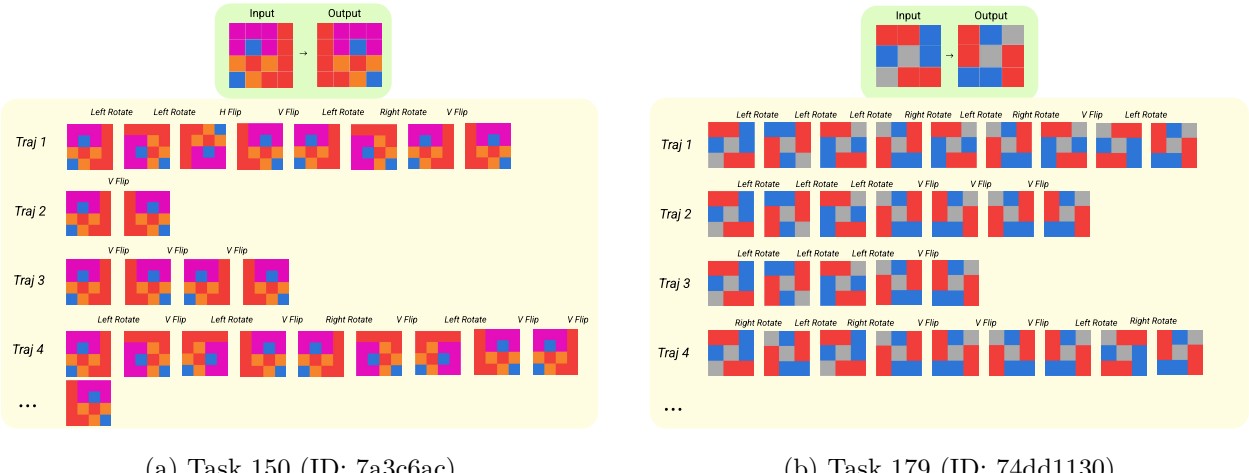

(a) Task 150 (ID: 7a3c6ac)               (b) Task 179 (ID: 74dd1130)

Figure 7: Visualization of generated trajectories for two ARC tasks. Each trajectory represents a sequence of transformations applied to the input grid to achieve the correct output. Task 150 (a) and Task 179 (b) demonstrate diverse action sequences, including rotations and flips, leading to successful solutions.

### 4.2.1 RQ1. Can reward functions informed by human priors promote human-aligned reasoning patterns?

To investigate whether human-inspired reward signals can effectively shape the structure of generated trajectories, we evaluate five reward models, each designed to encode specific reasoning heuristics:

- `Base`: Sparse reward only (no inductive bias).

- `Splus`: Emphasizes the Submit action.

- `Cycle`: Penalizes repeated actions (loop avoidance).

- `DF`: Discount factor encourages early goal completion.

- `TLR`: Penalizes long trajectories (conciseness).

To isolate the pure effect of different reward components, these experiments were conducted on the standard Geometric-Categorical (G-C) policy using the base reward scale (a reward of 15 for success).

**Results and Analysis**   We first analyze the effects of individual reward models (Tables 3 and 4). The Base model performs poorly, with only 5.26% unique success and reward entropy of 0.29, reflecting ineffective trajectory exploration. Among single reward settings, the model achieves the highest total success rate (**89.33%**) but with relatively low diversity (1.04), suggesting that its strong preference for short paths leads to early convergence on a narrow solution set.

`TLR` achieves lower total accuracy (67.33%) but the highest reward entropy (**1.08**), producing the most structurally diverse successful trajectories (**50** in Table 3). This suggests that `TLR` allows for a wider range of valid solution strategies while still guiding the model toward goal-oriented reasoning.

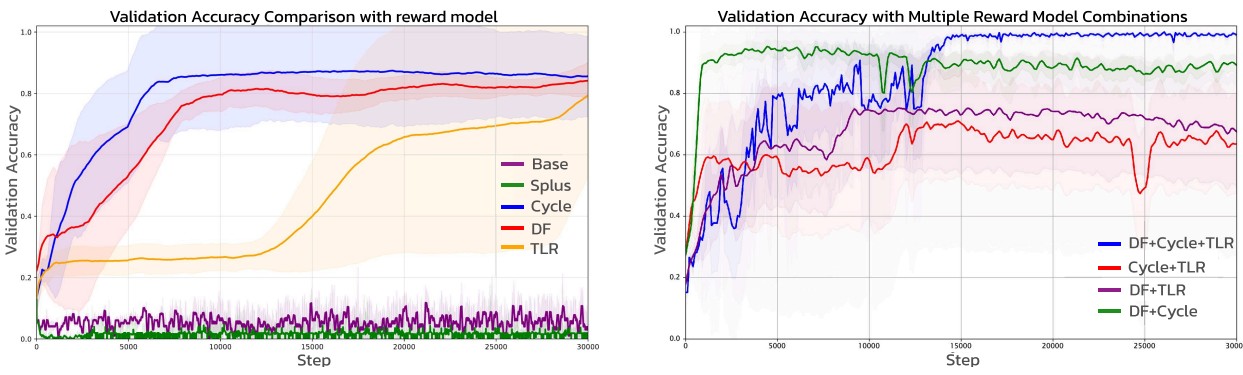

Figure 8: (Left) Validation accuracy for individual reward models. (Right) Validation accuracy for combined reward models.

`Cycle`, while reflecting the human tendency to avoid repetition, generates the largest number of unique trajectories (**100**), but with relatively low success (32 successful), indicating that diversity alone does not imply effective reasoning. Splus, despite its alignment with the task-specific "submit" action, shows limited impact (13.68% accuracy), implying that local task signals are insufficient without broader structural constraints.

We next examine reward combinations to assess whether structural biases interact synergistically (Tables 5, 6). `DF + Cycle + TLR` achieves perfect total success (**100%**), but with reduced diversity (entropy = 1.81), suggesting a strong convergence effect. `DF + TLR`, in contrast, yields a balance of high success (86%) and the highest diversity among all models (**2.96**), highlighting its effectiveness at preserving reasoning flexibility while maintaining performance.

| Reward | Total Unique Trajectories | Successful Unique Trajectories | Val_ACC (%) |
|--------|---------------------------|--------------------------------|-------------|
| DF | 36 | **27** | **75.47** |
| Cycle | **100** | **32** | 32.00 |
| TLR | 79 | **50** | 63.03 |
| Base | 76 | 4 | 5.26 |
| Splus | 95 | 13 | 13.68 |

Table 3: Success rate based on unique trajectories for different reward models.

| Reward | Total Trajectories | Total Successful Trajectories | Val_ACC (%) | $D_{reward}$ |
|--------|--------------------|-------------------------------|-------------|--------------|
| DF | 300 | **268** | **89.33** | 1.04 |
| Cycle | 300 | 96 | 32.00 | 0.90 |
| TLR | 300 | 202 | 67.33 | **1.08** |
| Base | 300 | 15 | 5.00 | 0.29 |
| Splus | 300 | 39 | 13.00 | 0.56 |

Table 4: Success rate and reward diversity based on total trajectories.

Notably, combinations that include `Cycle` often lead to more constrained trajectory spaces. For example, `Cycle + TLR` achieves 89% success but with the lowest diversity (0.92), reinforcing that while repetition avoidance is important, overly rigid enforcement may limit the model's exploration. These results support the notion that reward combinations do not simply stack effects but influence trajectory patterns in complex and often non-linear ways.

| Reward Combination | Unique Trajectories | Successful Unique Trajectories | Val_ACC (%) |
|--------------------|---------------------|--------------------------------|-------------|
| DF + Cycle + TLR | 5 | 5 | **100.00** |
| DF + TLR | 35 | 26 | 74.29 |
| Cycle + TLR | 38 | 29 | 76.32 |
| DF + Cycle | 47 | 29 | 61.70 |

Table 5: Success rate and reward diversity based on unique trajectories for reward combinations.

| Reward Combination | Total Trajectories | Successful Total Trajectories | Val_ACC (%) | $D_{reward}$ |
|--------------------|--------------------|-------------------------------|-------------|--------------|
| DF + TLR | 100 | 86 | 86.0 | **2.96** |
| DF + Cycle + TLR | 100 | 100 | **100.0** | 1.81 |
| Cycle + TLR | 100 | 89 | 89.0 | 0.92 |
| DF + Cycle | 100 | 91 | 91.0 | 1.50 |

Table 6: Success rate and reward diversity based on total trajectories for reward combinations.

To assess the marginal utility of the cycle constraint and to more deeply understand how different reward components affect structural efficiency, we analyzed the total number of cycle occurrences and the average trajectory length for various reward models. The results are presented in Table 7.

| Reward Model | Total Cycle Occurrences | Average Trajectory Length |
|--------------|-------------------------|---------------------------|
| Base | 408 | 9.59 |
| Cycle | 308 | 8.70 |
| DF + TLR | 255 | **8.12** |
| DF + Cycle | **246** | 8.15 |

Table 7: Comparison of reward models on trajectory efficiency metrics. Total Cycle Occurrences measures redundancy, while Average Trajectory Length measures brevity. Lower is better for both.

The results in Table 7 reveal the distinct effects of different reward strategies on improving trajectory efficiency. Both approaches that add constraints—either on structure (`DF + Cycle`) or length (`DF + TLR`)—significantly improve upon the `Base` and `Cycle`-only models.

Notably, the `DF + Cycle` model proved to be the most effective at reducing structural redundancy, achieving the lowest total number of cycle occurrences (246). On the other hand, the `DF + TLR` model produced the shortest average trajectories (8.12), making it the most effective in terms of overall brevity. This presents an interesting trade-off. Given that the primary goal of this work is to reduce inefficient reasoning paths, the `DF + Cycle` model, which most effectively minimizes cyclic (i.e., redundant) actions, can be considered the most successful in improving the logical quality of the trajectories. This result clearly demonstrates the marginal utility of an explicit cycle constraint over relying on length penalties alone to indirectly reduce redundancy.

This detailed analysis also puts the model's overall performance into perspective. As observed in Table 7, the average trajectory lengths for our optimized models are approximately 8 steps. This contrasts with human-provided solutions for similar ARC tasks, which often require fewer than 4 steps on average (Kim et al., 2025b). This significant gap underscores the ongoing challenge and importance of reward shaping in steering models toward more concise, human-aligned reasoning structures.

**Implications** These findings suggest that well-designed reward functions not only affect trajectory success but also influence the underlying structure and interpretability of the reasoning process. Individual rewards serve as proxies for distinct human problem-solving heuristics, and their combinations can shape exploration in either beneficial or restrictive ways.

While more constraints can increase success, they may also reduce diversity and generalization. For instance, the `DF + TLR` model demonstrates a favorable trade-off between structure and flexibility, whereas overly rigid combinations like `DF + Cycle + TLR` lead to narrower solution modes. These results support the idea that reward shaping is not merely about scoring optimization, but about modeling the reasoning biases that underlie human-like solution strategies.

This tension between success-driving constraints and diversity-promoting flexibility is central to our investigation. To dissect this trade-off more granularly and challenge the assumption that shorter paths are always superior, the following section provides a deep dive into the most critical parameter governing this balance: the maximum trajectory length, which dictates the model's fundamental exploration depth.

### 4.2.2 The Role of Exploration Depth: Challenging the "Shorter is Better" Heuristic

The preceding analysis of reward functions highlights the importance of trajectory length in guiding the GFlowNet. While mechanisms like `DF` and `TLR` encourage conciseness, a crucial question remains: what is the optimal exploration depth? To investigate this, and to challenge the simple assumption that shorter trajectories are always superior, we conducted a detailed analysis of the impact of varying the maximum episode length.

**Experiment Setup** The action space was fixed at 10 actions, and we varied the episode Length to observe how each setting influenced the model's performance. The episode Lengths tested were 4, 5, 10, 20, and 50. This setup allowed us to assess how different step limitations impact solution quality, exploration diversity, and learning efficiency.

**Results and Analysis** Our findings reveal a non-monotonic relationship between trajectory length and performance, as shown in Figure 9. While very short episodes (e.g., 4 or 5 steps) severely limited the model's ability to explore and resulted in low success rates, as detailed in Table 9, performance peaked at a moderate episode length of 20. This optimal length achieved the highest validation accuracy and a 97% total success rate. However, further increasing the exploration depth to 50 steps proved detrimental, causing the learning process to become unstable and fail entirely. This demonstrates that an overly large search space can dilute the learning signal, preventing the model from converging on valid solutions.

As shown in Table 8, this performance peak at an episode length of 20 also coincides with the highest diversity of solutions. The model generated 74 unique trajectories, 71 of which were successful, and exhibited the

highest reward distribution diversity ($D_{\mathrm{reward}} = 2.14$). This indicates that a sufficient, but not excessive, exploration depth is essential for discovering a wide variety of high-quality solutions.

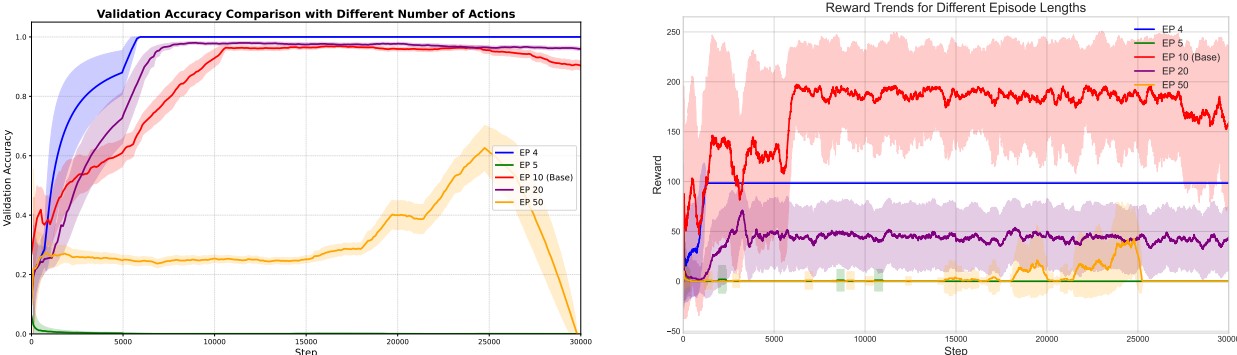

Figure 9: (Left) Performance comparison across episode length, showing the impact of action count on validation accuracy over training steps. (Right) Reward comparison: comparison of reward values over training steps, highlighting differences in reward trends.

| File | Unique Trajectories | Successful Unique Trajectories | Val_ACC (%) | D_reward |
|------|---------------------|-------------------------------|-------------|----------|
| EP 4 | 1 | 1 | 100.00 | 0.0 |
| EP 5 | 76 | 4 | 5.26 | 0.20 |
| EP 10 | 56 | 40 | 71.43 | 0.88 |
| EP 20 | 74 | 71 | 95.95 | 2.14 |
| EP 50 | 1 | 0 | 0.00 | 0.0 |

Table 8: Trajectory success rates and reward distribution diversity for various episode lengths and reward settings.

| File | Total Trajectories | Successful Total Trajectories | Val_ACC (%) |
|------|--------------------|-------------------------------|-------------|
| EP 4 | 100 | 100 | 100.0 |
| EP 5 | 100 | 5 | 5.0 |
| EP 10 | 100 | 59 | 59.0 |
| EP 20 | 100 | 97 | 97.0 |
| EP 50 | 100 | 0 | 0.0 |

Table 9: Total success rates for different episode lengths and reward configurations.

**Implications**   These results provide a crucial insight: the highest performance and diversity are achieved not through strict length minimization, but at a moderate trajectory length that balances efficiency with a sufficient 'exploration budget.' This finding directly informs our interpretation of the results from RQ1, showing that length-penalizing rewards should be seen as guides against inefficiency rather than tools for absolute minimization. Furthermore, it sets a crucial context for the following analysis of our Geometric policy, highlighting that its efficiency bias is most powerful when it does not prematurely curtail necessary exploration.

### 4.2.3   RQ2. Does using a Geometric distribution in the forward policy enhance efficiency in diverse trajectory generation?

We compare two GFlowNet variants that differ only in the choice of forward policy distribution: the original version using a Categorical distribution, and the revised version using a Geometric distribution. In both cases,

the backward policy ($P_B$) remains Categorical. To provide a clean baseline comparison of the policies' innate exploratory behaviors, both variants were trained with the simple sparse (Base) reward function at the base reward scale. This setup allows us to evaluate the pure effect of the policy distribution, without confounding factors from a complex reward structure.

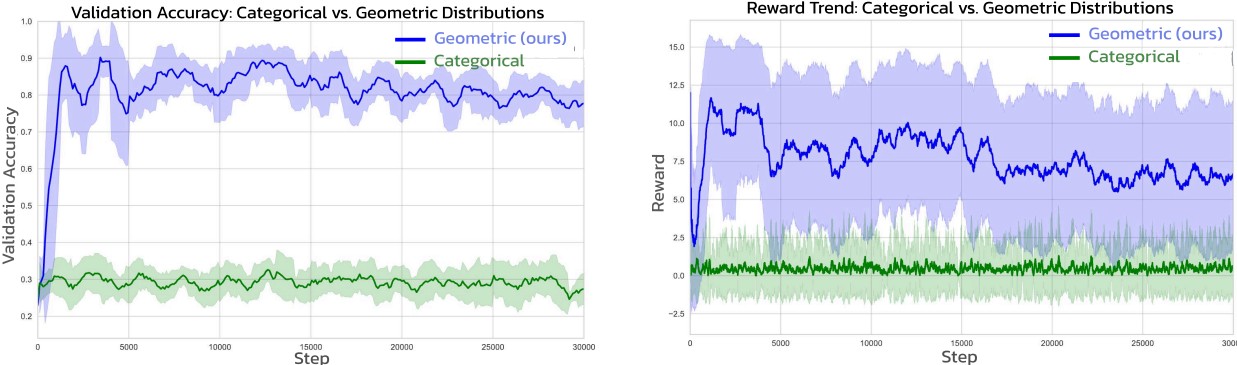

Figure 10: (Left) Validation accuracy comparison between Categorical (green) and Geometric (blue) distributions. (Right) Reward profiles comparison over training steps.

| Distribution | Unique Trajectories | Successful Unique Trajectories | Val_ACC (%) |
|---|---|---|---|
| Categorical | 75 | 10 | 13.33 |
| Geometric | 45 | **36** | **80.00** |

Table 10: Success rate based on unique trajectories for Categorical and Geometric distributions.

| Distribution | Total Trajectories | Successful Total Trajectories | Val_ACC (%) | $D_{reward}$ |
|---|---|---|---|---|
| Categorical | 100 | 28 | 0.28 | **3.38** |
| Geometric | 100 | **89** | 0.89 | 0.88 |

Table 11: Success rate based on total trajectories for Categorical and Geometric distributions.

**Results and Analysis**    The Geometric PF GFlowNet introduces an inductive bias toward concise and decisive action sequences—an intuition derived from analyzing human trajectory patterns, particularly in structured reasoning tasks like ARC. Unlike the original Categorical forward policy, which samples actions uniformly, the Geometric distribution prioritizes earlier actions, naturally biasing the model toward shorter trajectories.

As shown in Figure 10, the Geometric PF GFlowNet consistently achieves higher validation accuracy throughout training. Table 10 confirms that, while the Categorical policy produces a larger number of unique trajectories (**75** vs. 45), it leads to significantly fewer successful outcomes (10 vs. **36**). Likewise, Table 11 shows that the Geometric model outperforms in overall success rate (**89%** vs. 28%), despite exhibiting lower reward diversity ($D_{reward} = 0.88$ vs. **3.38**).

This contrast highlights that, although the Categorical policy exhibits broader surface-level exploration, much of it fails to reach valid solutions. In other words, diversity without effectiveness may not equate to useful reasoning. In contrast, the trajectories generated by the Geometric PF GFlowNet are more likely to be both correct and structurally sound. Notably, successful trajectories under the Geometric policy typically reach the goal within 2 to 4 steps—well below the episode limit of 10—demonstrating the model's inherent preference for efficient reasoning paths.

These paths exhibit key characteristics commonly associated with human problem-solving behavior: minimal reversals, consistent transformation patterns, and clear subgoal alignment. Although we do not claim

full cognitive equivalence, this emergent structure indicates a meaningful alignment between the model's exploratory behavior and human-like reasoning strategies.

**Implications** These results suggest that leveraging distributional biases—such as those imposed by the Geometric forward policy—can steer solution augmentors toward generating concise and interpretable reasoning paths. While the diversity of the Categorical model appears greater numerically, the Geometric PF GFlowNet better balances success and efficiency, especially in sparse-reward settings like ARC.

Moreover, the fact that this model can discover multiple valid reasoning paths for ARC-179—a task solvable in just a few steps—suggests that even simple inductive biases can facilitate structured solution generation. When combined with human-inspired reward functions (Section 4.2.4), this opens the possibility of extending the framework to more complex reasoning domains. Rather than replicating human reasoning in its entirety, the proposed approach reflects an early but promising step toward learning structural features of expert reasoning—such as brevity, decisiveness, and goal alignment—and embedding them within generative solution frameworks.

### 4.2.4 RQ3. Why does a Geometric forward policy excel in goal-conditioned reasoning tasks?

Having established the superior G-C policy (in Section 4.2.3) and identified effective reward components (`DF + Cycle` in Section 4.2.1), we now evaluate their final, combined effect. For this main experiment, we compare all policy combinations using the `DF + Cycle` reward scheme with the 15x amplified reward scale (as justified in Appendix B.1). We evaluate different combinations of forward ($P_F$) and backward ($P_B$) policies:

- **G-C:** Geometric for $P_F$ and Categorical for $P_B$.

- **G-G:** Geometric for both $P_F$ and $P_B$.

- **C-C:** Categorical for both $P_F$ and $P_B$.

- **C-G:** Categorical for $P_F$ and Geometric for $P_B$.

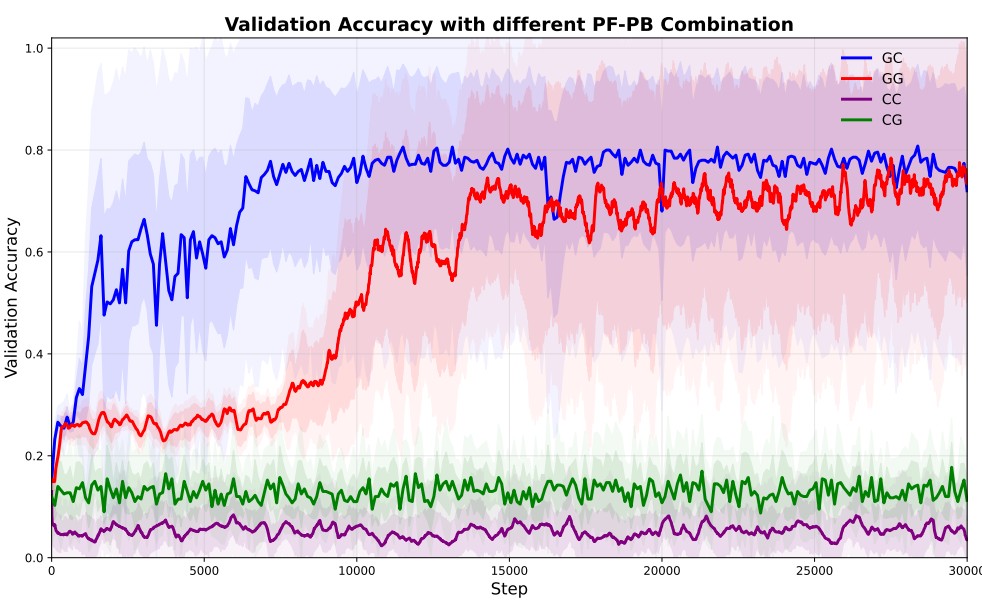

Figure 11: Validation accuracy for different $P_F$-$P_B$ combinations.

| $P_F$-$P_B$ | Unique Trajectories | Successful Unique Trajectories | Val_ACC (%) |
|---|---|---|---|
| G-G | 21 | 18 | **85.71** |
| G-C | 56 | **49** | **87.50** |
| C-G | 59 | 10 | 16.95 |
| C-C | **75** | 10 | 13.33 |

Table 12: Unique trajectory performance for different $P_F$-$P_B$ combinations.

| $P_F$-$P_B$ | Total Trajectories | Successful Total Trajectories | $D_{reward}$ |
|---|---|---|---|
| G-G | 100 | **97** | **2.15** |
| G-C | 100 | **97** | **2.09** |
| C-G | 100 | 11 | 0.73 |
| C-C | 100 | 10 | 3.39 |

Table 13: Overall success rates for different $P_F$-$P_B$ combinations.

**Results and Analysis**  We compare four configurations of forward ($P_F$) and backward ($P_B$) policies. Both G-G and G-C achieve the highest total success rate (97.0%), but the G-C pairing yields a significantly higher number of successful unique trajectories (49 vs. 18), indicating better structural generalization. C-G and C-C perform poorly, both in accuracy and trajectory success.

This pattern suggests that the forward and backward policies contribute differently to learning dynamics. The forward policy ($P_F$) is responsible for generating exploratory trajectories, while the backward policy ($P_B$) adjusts the reverse probabilities to match the desired flow based on reward signals. When both policies are heavily biased (e.g., G-G), the backward policy may no longer act as a corrective mechanism. Instead, it may reinforce the forward bias, narrowing the trajectory space and reducing diversity.

In contrast, the G-C setup preserves high success while also producing a wider range of successful solutions. This indicates that an asymmetric policy design—where $P_F$ is biased for efficient exploration and $P_B$ remains uniform to stabilize flow alignment—achieves better reasoning diversity without sacrificing performance. The observed $D_{reward}$ values support this: G-G has slightly higher entropy (2.15) than G-C (2.09), but the diversity of correct solutions is much higher in G-C.

We also observe that C-C has the highest reward entropy (3.39) but fails to achieve meaningful performance, indicating that reward distribution diversity alone does not imply reasoning quality. High entropy may reflect exploratory breadth, but it must be interpreted alongside success metrics. Ultimately, reward diversity without trajectory effectiveness may signal unstructured exploration, not reasoning robustness.

**Insights**  These findings reinforce the importance of decoupling exploration and correction roles in bidirectional learning. The forward policy drives goal-directed trajectory generation, while the backward policy should facilitate stable reward-aligned flow learning. We hypothesize that if both policies impose strong, aligned biases (as in G-G), the resulting feedback loop can distort the reward structure and impair the model's ability to generalize across diverse solutions.

Moreover, a successful unique trajectory count emerges as a valuable proxy for structural reasoning diversity. Models such as G-C not only succeed frequently but also do so via a wider variety of valid solution paths. This suggests that asymmetric policy pairing enables the model to learn broader reasoning structures—a key capability in ARC-like tasks where flexible generalization is crucial.

Finally, reward distribution diversity, as captured by entropy, offers useful information about the model's exploratory behavior, but does not alone indicate reasoning effectiveness. Reward entropy should be interpreted in conjunction with success metrics to evaluate the utility—not just the scope—of the model's exploration.

### 4.3 Downstream Evaluation on Large Language Models

To assess whether GFlowNet-generated trajectories provide tangible utility in reasoning models, we conduct a downstream evaluation using LLM. Our objective is to determine whether exposure to these synthetic trajectories can teach an LLM to reason through ARC-style tasks using DSL-based action sequences.

**Experimental Setup.** We fine-tune `LLaMA 3.1 8B Instruct`, a powerful language model, on approximately 10k GFlowNet-generated trajectories per ARC task. The target output is the complete action sequence required to transform the input into the correct output. This is a stricter evaluation than final output prediction, as it tests whether the model has internalized procedural reasoning in DSL format.

**Dataset and Tasks.** We select 7 ARC tasks of varying difficulty and structure, covering geometric transformations, color-based logic, and compositional reasoning. Each task contains a mixture of high-level and low-level operations. For each task, 100k trajectories were generated, and **10k were sampled uniformly for fine-tuning**.

**Results.** The results in Table 14 demonstrate a clear performance improvement. The baseline model fails all tasks, generating only natural language or malformed DSL. The fine-tuned model, however, correctly solves two tasks and produces valid DSL in several others. Qualitatively, the model demonstrates awareness of transformation steps such as rotation, reflection, and conditional masking.

Table 14: Downstream performance of `LLaMA 3.1 8B Instruct` on ARC tasks before and after fine-tuning with GFlowNet-generated trajectories.

| Task # (ID) | Baseline (Pretrained) | Fine-Tuned w/ GFlowNet Aug. |
|:---:|:---:|:---:|
| 87 (3c9b0459) | ✗ | ✗ |
| 140 (6150a2bd) | ✗ | ✓ |
| 150 (67a3c6ac) | ✗ | ✗ |
| 155 (68b16354) | ✗ | ✗ |
| 179 (74dd1130) | ✗ | ✓ |
| 241 (9dfd6313) | ✗ | ✗ |
| 380 (ed36ccf7) | ✗ | ✗ |
| **Overall Accuracy** | **0 / 7 (0%)** | **2 / 7 (28.6%)** |

**Qualitative Insights.** The fine-tuned model learns to generate syntactically valid DSL expressions such as `[rotate, xor, color_mask, submit]` rather than vague or irrelevant text (e.g., "Let's analyze the pattern..."). This indicates that the model acquired procedural understanding rather than just output matching. Notably, the two successful tasks involve spatial transformations, suggesting that GFlowNet augmentation helped capture geometric reasoning structures.

**Implications.** These findings provide strong proof-of-concept evidence that GFlowNet-generated trajectories can serve as adequate supervision for training reasoning-capable LLMs. Even in this limited experiment, trajectory-level augmentation enables the LLM to generalize in domains where it previously failed. This supports the broader thesis that generating diverse, structured reasoning traces can improve model generalization and interpretability.

### 4.4 Comparison to Trajectory Augmentation Baseline

While many augmentation strategies exist for input–output pairs in vision and NLP tasks, there is no established baseline for augmenting *solution trajectories* in ARC. Existing efforts such as ReARC (Hodel, 2024) primarily augment the *input grids*, associating each augmented instance with a single canonical solution. However, this style of augmentation does not capture the diversity of valid reasoning paths that can lead to the same outcome. Consequently, we adopt ReARC-style augmentation as a proxy baseline for comparison.

**Experimental Setup.** To fairly evaluate the utility of our framework, we compared LLMs fine-tuned on two distinct forms of augmented data, while keeping all model and training configurations fixed. Both models used the **LLaMA 3.1 8B Instruct** backbone and were fine-tuned with identical LoRA settings (r=16, $\alpha = 32$).

- **ReARC-style Augmentation:** ~70,000 examples were generated via input grid perturbations, each paired with a single canonical solution sequence (e.g., a rotation or reflection). This provides high variability in the input space but minimal variability in solution paths.

- **GFlowNet-based Augmentation:** ~10,000 trajectories were synthesized across 7 ARC problems using our framework. Unlike the baseline, these data contain multiple diverse trajectories per task, better reflecting the exploratory nature of ARC reasoning.

**Results.** The GFlowNet-augmented model solved **2 of 7 tasks (28.6%)**, specifically Problem 140 (6150a2bd) and Problem 179 (74dd1130). By contrast, the ReARC baseline solved only **1 task (14.3%)**, despite being trained on seven times more examples. Moreover, the baseline consistently produced the same action sequence in 5 of 7 tasks, indicating severe overfitting to canonical solutions. In contrast, the GFlowNet-trained model generated a more diverse set of valid sequences, enabling it to generalize to problems requiring geometric transformations.

Table 15: Downstream performance of `LLaMA 3.1 8B Instruct` on ARC tasks with ReARC-style vs. GFlowNet-based augmentation.

| Task # (ID) | ReARC-style Aug. | GFlowNet Aug. |
|:---:|:---:|:---:|
| 87 (3c9b0459) | ✗ | ✗ |
| 140 (6150a2bd) | ✓ | ✓ |
| 150 (67a3c6ac) | ✗ | ✗ |
| 155 (68b16354) | ✗ | ✗ |
| 179 (74dd1130) | ✗ | ✓ |
| 241 (9dfd6313) | ✗ | ✗ |
| 380 (ed36ccf7) | ✗ | ✗ |
| **Overall Accuracy** | **1 / 7 (14.3%)** | **2 / 7 (28.6%)** |

**Analysis.** This experiment shows that diversity in *solution paths* is more valuable than diversity in input grids. While the ReARC baseline introduced many surface-level variations, its reliance on a single fixed solution sequence limited both its learning signal and generalization. In contrast, our GFlowNet framework, even with significantly fewer examples, exposed the model to rich intermediate states and trajectory structures, enabling it to learn the *process* of problem solving rather than memorizing outcomes.

**Conclusion.** This controlled comparison highlights that **trajectory-level augmentation**, which enriches the diversity of solution paths, can outperform input-driven baselines in ARC. By generating diverse and meaningful solution trajectories, our framework doubled downstream accuracy compared to the canonical augmentation approach, despite using fewer data. These findings strengthen the case for trajectory-level augmentation as a more effective training signal for reasoning-intensive tasks such as ARC.

## 5 Conclusion

In this study, we proposed a novel GFlowNet-based framework for solution augmentation in ARC problems. By leveraging a Geometric forward policy, a Categorical backward policy, and human-inspired reward models, our method discovers structurally efficient and diverse solution trajectories in sparse-reward environments.

### 5.1 Summary and Implications

Our experiments demonstrate that:

- **Policy and Reward Design:** The combination of a Geometric forward policy ($P_F$) with human-inspired reward shaping (discounting, length regularization, and cycle penalties) substantially improves success rates over categorical baselines, showing that efficiency-biased heuristics and structural guidance are critical for effective learning.

- **Exploration–Policy Trade-off:** Our analysis reveals that the best performance and diversity emerge at moderate trajectory lengths, not strict minimization. Moreover, pairing a Geometric $P_F$ with a Categorical backward policy ($P_B$) balances directed exploration and flow correction, yielding both high accuracy and diverse successful trajectories.

- **Downstream Validation:** Fine-tuning LLaMA 3.1 8B with our GFlowNet-generated trajectories led to a significant improvement ($0\% \rightarrow 28.6\%$). In a controlled experiment, our 10k diverse trajectories outperformed 70k ReARC-style input-augmented examples (28.6% vs. 14.3%), highlighting the superiority of trajectory diversity over mere input augmentation.

- **Efficiency and Test-Time Augmentability:** Our framework can generate over 100k trajectories per task in under an hour on a single GPU (2,000/min). This logarithmic yield of unique solutions enables efficient augmentation without redundancy and makes real-time, test-time augmentation feasible.

- **Practical Significance:** Rather than claiming cognitive alignment, our findings show that GFlowNet can be guided toward syntactic features of efficient problem solving—brevity, structure, and diversity—through policy and reward design. This positions GFlowNet as a scalable, expert-aligned solution generator for reasoning-intensive tasks like ARC.

Rather than claiming full cognitive alignment, our work shows that GFlowNet can be guided to reflect aspects of efficient problem-solving—such as brevity and structural clarity—through reward design. These findings suggest that carefully structured inductive biases (i.e., policy and reward) can guide GFlowNet toward expert-aligned solution augmentation, highlighting its utility for complex reasoning tasks such as ARC.

### 5.2 Limitations and Future Work

While our proposed framework yields promising results, several limitations remain:

- **On-Policy Training Sensitivity:** The current approach relies on on-policy training, which is sensitive to the quality of initial trajectory samples. Early poor samples may cause high variance in learning. Future work will explore hybrid training schemes that combine the robustness of off-policy updates with the stability of on-policy guidance.

- **Restricted Task Scope:** This study focuses on ARC tasks involving full-grid transformations. Extending the framework to handle tasks that require localized sub-grid manipulations poses a significant challenge due to the exponentially growing action space. We plan to address this by leveraging ARCLE (Lee et al., 2024) and developing submodules capable of supporting fine-grained selection and transformation operations.

- **Adaptive Exploration Depth and Reward Weighting**: Our findings highlight the critical role of episode length. The current framework uses a fixed length, but future work should explore adaptive methods for determining the optimal exploration depth dynamically based on task complexity. Furthermore, as a natural extension, exploring adaptive reward weighting and training-phase scheduling could help to dynamically manage the trade-offs between different reasoning heuristics in more complex or open-ended domains.

### 5.3 Final Remarks

Our work demonstrates the potential of GFlowNet as a powerful mechanism for automated solution augmentation in reasoning-intensive tasks. By incorporating human-inspired reward structures and asymmetric policy design, we guide the model to generate concise, diverse, and high-quality solutions—an essential capability in sparse-reward environments.

Rather than functioning purely as a solver, GFlowNet here serves as an expert-aligned **solution generator**. This opens opportunities for integrating our approach into broader reasoning pipelines, where diverse solution candidates are required for further verification, ranking, or learning. Our framework's scalability and downstream effectiveness position it as a practical component in such pipelines, especially under compute-limited test-time settings. We believe that future advances in exploration strategies, structural bias modeling, and downstream integration will further enhance the applicability of GFlowNet to System-2-level tasks, ultimately contributing to more general and interpretable artificial intelligence systems.

## Acknowledgements

This work was supported by NRF (Reinforcement Learning-Based Program Synthesis Techniques for Solving Abstract Visual Reasoning Problems by Identifying Patterns and Combining Concepts; RS-2024-00451162, Developing Abstraction and Reasoning Capability for AI Model; RS-2024-00454000), IITP (Enhancing AI Model Reliability Through Domain-Specific Automated Value Alignment Assessment; RS-2024-00445087, Development of Artificial Complex Intelligence for Conceptually Understanding and Inferring like Human; RS-2023-00216011), and GIST (KH0330, Future-leading Specialized Research Project) grants funded by the Ministry of Science and ICT, Korea.

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

# A Extended Background Details

## A.1 GFlowNet Training Mechanism and Flow Matching

**Flow Matching Condition.** GFlowNets maintain a flow $F(s' \to s)$ for each directed edge $(s', s)$ in a Directed Acyclic Graph (DAG), ensuring no probability mass is lost or gained within each state:

$$\sum_{s'} F(s' \to s) \;=\; \sum_{s''} F(s \to s''), \tag{10}$$

where $F(s' \to s)$ is the flow from state $s'$ to $s$. This condition (Equation 10) ensures that each state's inflow equals its outflow. In practice, $F(s' \to s)$ is factored through a forward policy $P_F(s \mid s')$.

**Trajectory Balance (TB) Loss.** Building on Flow Matching, the TB loss (Malkin et al., 2022) ensures a global consistency between forward and backward paths:

$$Z \prod_{t=1}^{n} P_F(s_t \mid s_{t-1}) \;=\; R(x) \prod_{t=1}^{n} P_B(s_{t-1} \mid s_t), \tag{11}$$

where $Z$ is a trainable constant and $R(x)$ the reward function for trajectory $x$. Minimizing:

$$\mathcal{L}_{\text{TB}}(\theta) \;=\; \left[ \log Z_\theta \;+\; \sum_{t=1}^{n} \log P_F(s_t \mid s_{t-1}; \theta) \;-\; \log R(x) \;-\; \sum_{t=1}^{n} \log P_B(s_{t-1} \mid s_t; \theta) \right]^2 \tag{12}$$

aligns the probability of forward-sampled trajectories with their corresponding reward-proportional flows.

**Training Steps.** We outline the GFlowNet training procedure:

1. **Forward Sampling:** From initial state $s_0$, sample actions $a_t \sim P_F(\cdot \mid s_{t-1})$. Collect the resulting trajectory $x = (s_0, \ldots, s_n)$.

2. **Reward Computation:** Evaluate $R(x)$, e.g. 1 if it solves an ARC puzzle, else 0.

3. **Backward Sampling:** Use $P_B$ to reconstruct or partially revisit states from $s_n$ to $s_0$.

4. **TB Loss Computation:** Compute $\mathcal{L}_{\text{TB}}$ via Equation equation 12.

5. **Parameter Update:** Optimize $\theta$ to minimize $\mathcal{L}_{\text{TB}}$, adjusting both $P_F$ and $P_B$ accordingly.

By iterating these steps, GFlowNets learn to focus on high-reward trajectories while maintaining a diverse distribution of solutions.

## A.2 Extended ARC Details

Although this work focuses on *whole-grid transformations*, many ARC puzzles require partial selection (e.g. coloring only a sub-region). This drastically increases the action space because selecting subsets of a $30 \times 30$ grid can be combinatorial. We note:

- **Dataset Complexity:** Some tasks have grid dimensions smaller than $30 \times 30$, but the upper limit still poses a challenge.

- **Potential Approaches for Partial-Selection:** We can adopt hierarchical policies that first *select* a region, then *transform* it. This approach can be integrated into GFlowNets by factoring the forward policy into multiple steps (Appendix of (Lee et al., 2024)).

- **Sparse Rewards and OOD Issues:** ARC test grids often deviate from training grids (different shapes, new patterns), demanding robust generalization. GFlowNets' ability to maintain multiple solutions is beneficial, but design of the reward function must handle rarely-seen corner cases.

**Reasoning Dimensions in ARC.** Following the framework of Lee et al. (2025), we consider three core cognitive dimensions of reasoning assessed by ARC:

- **Compositionality:** The ability to combine simple building blocks (e.g., functions or rules) into more complex transformations. ARC tasks often require models to compose multiple abstract operations.

- **Productivity:** The ability to extrapolate beyond observed examples and generate novel input-output mappings consistent with the inferred rules. ARC evaluates this by testing generalization to new input grids.

- **Logical Coherence:** The ability to maintain consistent internal logic throughout the reasoning process. This includes applying learned transformations in a rule-consistent manner across diverse contexts.

### A.2.1 Formal Problem Definition for Whole-Grid ARC

Formally, each ARC task is a function $f$ mapping $x \in \mathcal{X}$ to $y \in \mathcal{Y}$. We define a sequence of transformations $a_1, \ldots, a_T$ operating on the entire grid to produce $y$.

$$x^{(t+1)} = f\big(x^{(t)}, a_t\big) \quad \text{with} \quad x^{(0)} = x, \ \ x^{(T)} = y.$$

We use a binary reward $R(x) = 1$ if $x^{(T)}$ matches the target output $y$, else 0. Further details on Markov Decision Process (MDP) formulations, action definitions, and potential expansions can be found in (Lee et al., 2024).

## B Ablation Studies

The ablation studies focus on identifying which components of GFlowNet contribute most to its success in solving ARC problems. We evaluate the effects of reward scaling, different distribution combinations, and on-policy vs off-policy learning.

### B.1 Effect of Reward Scale

In this study, we analyze the impact of varying reward scales on GFlowNet's performance. By adjusting the magnitude of rewards, we observe changes in learning efficiency, trajectory diversity, and reward distribution, aiming to identify the optimal reward scale for effective solution generation.

**Experiment Setup** As described in the hyperparameter settings, the base rewards were set at 15 for correct solutions, 0 for incorrect ones, and 10 for solutions ending with the *submit* action without reaching the maximum episode length. To examine the effects of different reward scales, we conducted experiments with the following settings:

- **Low Scale**: Reduced base rewards to 3 (by dividing the base reward by 5) for correct solutions and 2 for submit-ended trajectories.

- **Moderate Scale (Base)**: Maintained the default base rewards of 15 for submit-ended trajectories, 10 for trajectories that ended due to the maximum episode limit, and 0 for incorrect trajectories.

- **High Scale**: Increased base rewards to 225 for correct solutions and 150 for submit-ended trajectories (by multiplying the base reward by 15).

- **Very High Scale**: Significantly increased base rewards to 1500 for correct solutions and 1000 for submit-ended trajectories (by multiplying the base reward by 100).

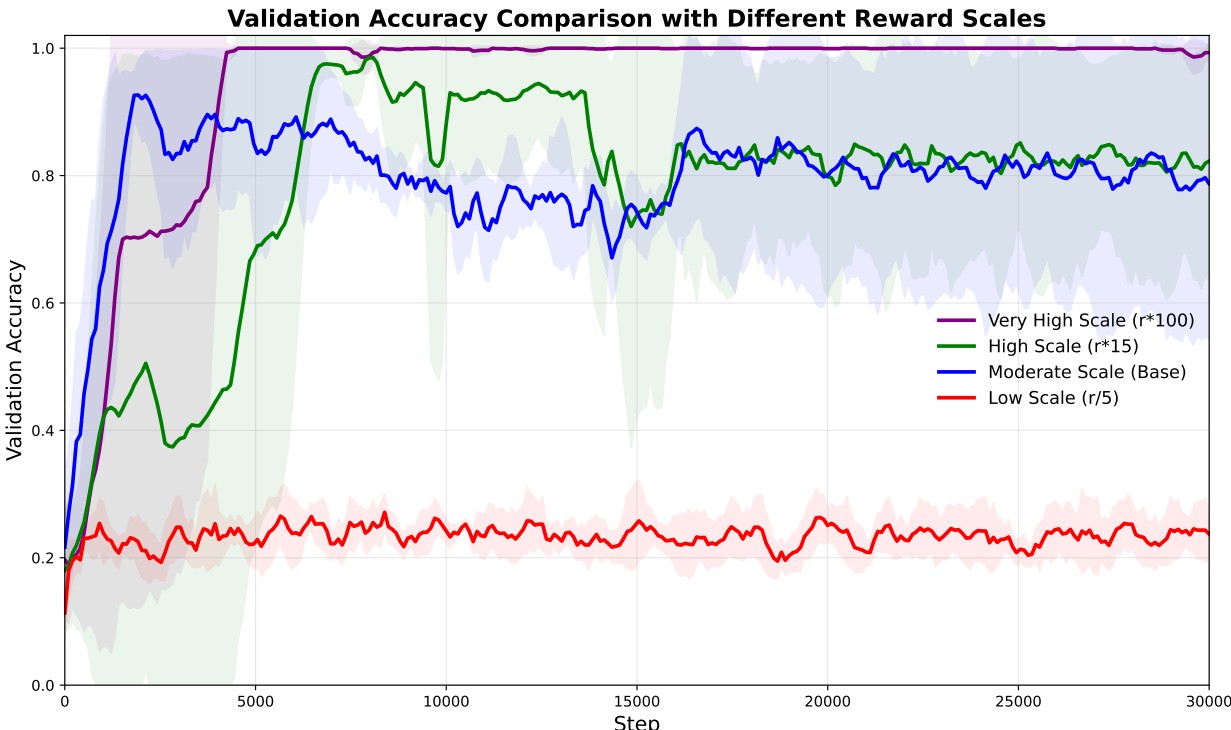

Figure 12: Performance comparison across different reward scales, showing the impact of reward values on validation accuracy over training steps.

**Results and Analysis**   Tables 16 and 17 summarize the effects of different reward scales on success rates for unique and total trajectories. A moderate reward scale (Base) demonstrated a balanced improvement in solution diversity and success rate, achieving a 71.43% success rate for unique trajectories and 59.0% for total trajectories.

The effect of excessively high scaling (e.g., $100r$) led to a 100% success rate for both unique and total trajectories. However, it also significantly reduced solution diversity, producing only a single unique trajectory. This outcome suggests that while very high rewards lead to success, they may limit exploration. To confirm this effect, we calculated the diversity metric $D_{\text{reward}}$, which showed a reduction in reward distribution diversity at very high rewards, supporting the conclusion that excessive scaling discourages exploration.

Conversely, a lower reward scale ($r/5$) resulted in no successful trajectories, as indicated by a 0% success rate for both unique and total trajectories. This lack of success demonstrates that low rewards do not sufficiently reinforce correct solutions, causing the model to struggle to distinguish high- from low-quality trajectories.

Figure 12 illustrates the learning curves for each reward setting, showing that lower reward scales lead to noticeably poorer initial performance due to insufficient reinforcement. In contrast, moderate scaling enables broader exploration of the solution space, allowing the model to generalize effectively across different ARC tasks.

Furthermore, the diversity metrics, including trajectory diversity $D_{\text{traj}}$ and reward distribution diversity $D_{\text{reward}}$, show that moderate reward scaling promotes greater exploration of successful trajectories. This finding suggests that a well-scaled reward signal enhances the model's ability to discover correct solutions while encouraging diverse trajectory exploration.

**Conclusion**   These findings suggest that while moderate reward scaling enables GFlowNet to balance efficient learning and exploration, excessively high rewards yield diminishing returns in terms of diversity. Although high rewards increase task completion rates, they may reduce exploration and solution diversity.

| Reward Scale | Unique Trajectories | Successful Unique Trajectories | Val_ACC (%) |
|---|---|---|---|
| Low Scale | 100 | 0 | 0.00 |
| Moderate Scale (Base) | 56 | 40 | 71.43 |
| High Scale | 74 | 53 | 71.62 |
| Very High Scale | 1 | 1 | 100 |

Table 16: Success rate and reward distribution diversity based on unique trajectories across different reward scales.

| Reward Scale | Total Trajectories | Successful Total Trajectories | Val_ACC (%) | D_reward |
|---|---|---|---|---|
| Low Scale | 100 | 0 | 0.0 | 0.5139 |
| Moderate Scale (Base) | 100 | 59 | 59.0 | 0.8817 |
| High Scale | 100 | 75 | 75.0 | 0.6739 |
| Very High Scale | 100 | 100 | 100.0 | 0.0 |

Table 17: Success rate based on total trajectories across different reward scales.

Conversely, low reward scales do not provide sufficient reinforcement for effective learning. Optimal reward scaling is thus essential for supporting both accuracy and diversity, meeting the ARC task's requirements for generalization and high-quality solutions.

Statistical tests (e.g., chi-squared and Fisher's exact tests) confirmed a significant difference in success rates and diversity metrics between moderate and very high reward scales, reinforcing that balanced reward scaling is critical for effective exploration and efficient learning in complex tasks.

### B.2 Action Number Performance Comparison

In this ablation study, we investigate the impact of varying the number of actions (3, 4, 5, and 10) on learning efficiency, exploration capacity, and trajectory diversity, further validating the insights obtained from the primary experiments.

**Experiment Setup** To solve ARC Task 179, which features a "Diagonal Flip" transformation, a minimum of three actions is required. While alternative solutions exist, the shortest solution involves three transformations: rotating the grid by 90 degrees, performing a horizontal flip, and executing the submit action. Starting with this minimal action set, we incrementally increased the number of actions to observe how each configuration affected validation accuracy and trajectory diversity over a maximum episode length of 10. This setup allowed us to explore the relationship between action granularity and the model's learning and exploration capabilities.

**Results and Analysis** As shown in Figure 13, models with three and five actions achieved high validation accuracy, with the three-action configuration reaching high performance relatively early in training. This indicates that a smaller action space allows more focused exploration, leading to efficient learning. Notably, the five-action configuration achieved near-optimal accuracy, outperforming both the four- and ten-action configurations in terms of learning efficiency. In contrast, the ten-action configuration, despite its expanded action space, failed to improve accuracy, suggesting that an excessive number of actions may impede effective exploration within the same training duration.

The reward trend graph further illustrates the differences across action configurations. The five-action setup shows stable and high rewards, indicating that this configuration balances action granularity with exploration capacity. The four-action configuration stabilizes at a moderate reward level after an initial exploration phase. In contrast, the ten-action configuration exhibits consistently low rewards, suggesting that GFlowNet struggles to optimize effectively in larger action spaces without additional guidance. This finding aligns with prior research, suggesting that overly large action or state spaces can dilute exploration, leading to suboptimal learning.

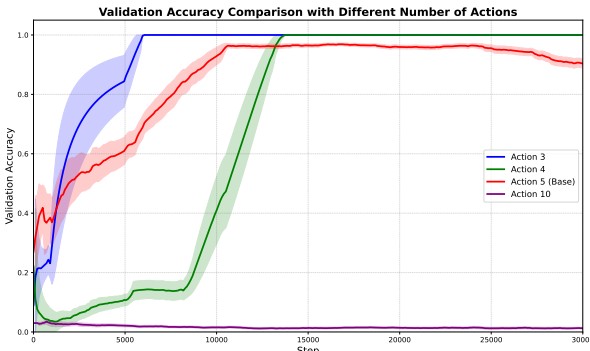 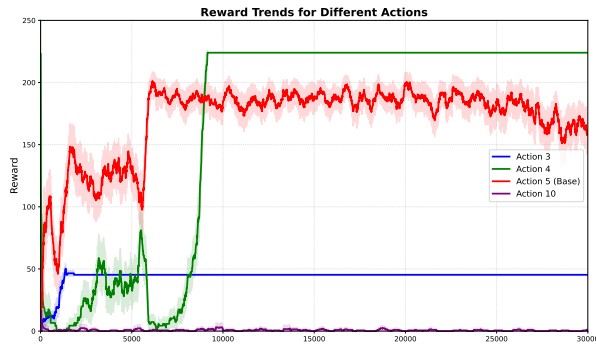

Figure 13: (Left) Performance comparison across number of actions, showing the impact of action count on validation accuracy over training steps. (Right) Reward comparison: comparison of reward values over training steps for each action configuration, highlighting differences in reward trends.

While expanding the action space increases trajectory diversity, it does not necessarily correlate with higher validation accuracy. For example, although the ten-action configuration showed greater trajectory diversity, it suffered from reduced learning efficiency and failed to achieve high rewards. This suggests that, while larger action spaces introduce more potential trajectories, they may also complicate exploration, especially without additional guidance.

The calculated *Reward Distribution Diversity* $D_{\mathrm{reward}}$ values provide further insight. As seen in Tables 18 and 19, $D_{\mathrm{reward}}$ for the three- and four-action configurations is zero, indicating uniform rewards due to limited trajectory diversity. In contrast, the ten-action configuration has a $D_{\mathrm{reward}}$ of 0.4439, and the five-action configuration yields 0.6739, reflecting greater reward diversity as the action space expands. Although increased diversity reflects broader exploration, it does not necessarily lead to improved performance, as shown by the ten-action configuration's low success rate and accuracy.

| File | Unique Trajectories | Successful Unique Trajectories | Val_ACC (%) | Total Trajectories | D_reward |
|---|---|---|---|---|---|
| $n(a) = 3$ | 1 | 1 | 100.00 | 100 | 0.0 |
| $n(a) = 4$ | 1 | 1 | 100.00 | 100 | 0.0 |
| $n(a) = 5$ (Base) | 41 | 26 | 63.41 | 100 | 0.6739 |
| $n(a) = 10$ | 95 | 11 | 11.58 | 100 | 0.4439 |

Table 18: Summary of unique and successful trajectories, success rates, total trajectories, and reward distribution diversity for each action configuration.

| File | Successful Total Trajectories | Val_ACC (%) | D_reward | Comments |
|---|---|---|---|---|
| $n(a) = 3$ | 100 | 100.0 | 0.0 | Minimal exploration with consistent success |
| $n(a) = 4$ | 100 | 100.0 | 0.0 | Limited diversity but high success rate |
| $n(a) = 5$ (Base) | 81 | 81.0 | 0.6739 | Balanced exploration and success |
| $n(a) = 10$ | 11 | 11.0 | 0.4439 | High diversity, low success rate |

Table 19: Total success rates, reward distribution diversity, and additional comments on trajectory diversity and performance for each action configuration.

The three-action and four-action configurations both achieved 100% success rates for unique trajectories, indicating that each generated trajectory successfully completed the task, though with limited diversity. Conversely, the ten-action configuration showed the highest number of unique trajectories (95) but had a low success rate (11.58%), suggesting that a large action space dilutes effective exploration, leading to lower task completion rates. The five-action configuration balanced exploration and success, yielding 41 unique trajectories with a success rate of 63.41%.

For the five- and ten-action configurations, statistical tests (chi-squared and Fisher's exact tests) confirmed a statistically significant difference in success rates, with the five-action configuration performing better among unique trajectories. This reinforces the idea that an optimal action space size is essential for effective learning.

These findings suggest that balancing action space size is critical for effective exploration and learning. Larger action spaces may require additional mechanisms, such as hybrid training approaches or managed replay buffers, to selectively sample high-quality data for focused exploration (Shen et al., 2023; Sendera et al., 2024). In this study, we implemented off-policy mechanisms to address this challenge. Future research could develop models capable of handling extensive search spaces more effectively, potentially by incorporating adaptive exploration techniques that adjust to the complexities of larger action spaces.

### B.3 Generated Trajectories and application to other tasks

To further analyze GFlowNet's trajectory generation and evaluate its applicability to other tasks, we conducted task-specific experiments on selected ARC tasks. These experiments aimed to examine the diversity of generated trajectories, their success rates, and the consistency of reward-based learning across different task types.

**Trajectory Visualization and Analysis**  Figure 14 illustrates multiple trajectories generated for two ARC tasks: Task 150 and Task 179. Each trajectory represents a unique sequence of transformations (e.g., rotations, flips) applied to the input grid to generate the correct output:

- **Task 150 (67a3c6ac):** The generated trajectories include efficient sequences (e.g., fewer steps) and redundant solutions with repeated actions. This demonstrates GFlowNet's exploration of diverse paths to the same output.

- **Task 179 (74dd1130)**: While successful trajectories are generated, Task 179 highlights the tendency of some solutions to include unnecessary repetitive actions, suggesting room for further optimization.

- These examples showcase GFlowNet's ability to discover a range of action sequences, balancing diversity and correctness in trajectory generation.

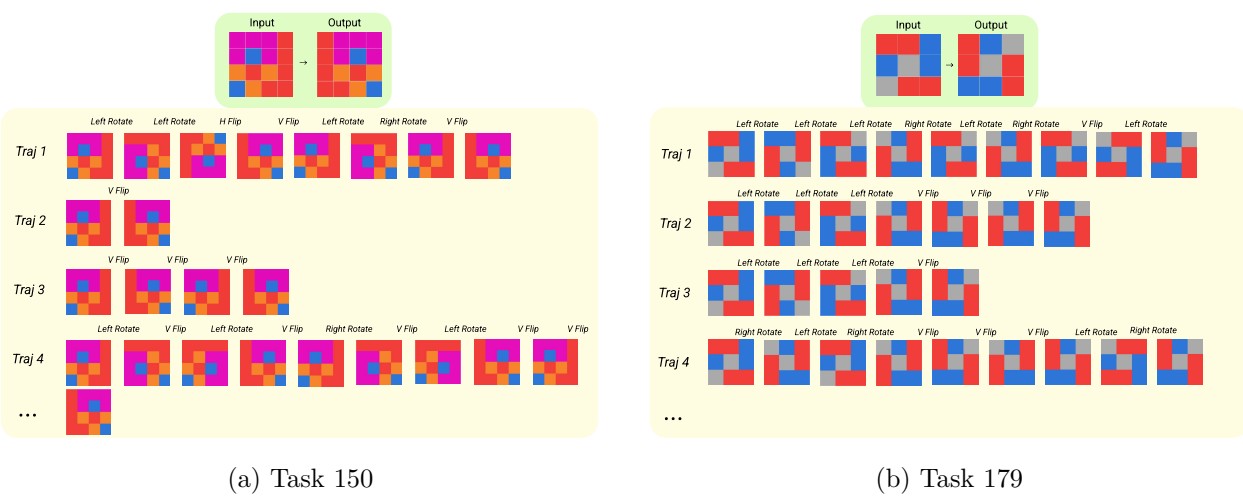

(a) Task 150  (b) Task 179

Figure 14: Visualization of generated trajectories for two ARC tasks. Each trajectory represents a sequence of transformations applied to the input grid to achieve the correct output. Task 150 (a) and Task 179 (b) demonstrate diverse action sequences, including rotations and flips, leading to successful solutions.

**Task-wise Results and Observations**  Table 20 summarizes the results of our task-specific analysis, including the number of unique trajectories, their success rates, and the reward diversity (entropy).

| Task # | Unique Trajectories (Total) | Successful Unique (Total) | Success Rate (Unique, Total, %) | Reward Diversity (Entropy) |
|---|---|---|---|---|
| 87 (3c9b0459) | 15 (300) | 15 (300) | 100.00 (100.00) | 0.72 |
| 140 (6150a2bd) | 3 (300) | 3 (300) | 100.00 (100.00) | 0.00 |
| 150 (67a3c6ac) | 100 (300) | 10 (30) | 10.00 (10.00) | 0.37 |
| 155 (68b16354) | 3 (300) | 3 (300) | 100.00 (100.00) | 0.00 |
| 241 (9dfd6313) | 105 (300) | 99 (280) | 94.29 (93.33) | 0.37 |
| 380 (ed36ccf7) | 3 (300) | 3 (300) | 100.00 (100.00) | 0.00 |

Table 20: Task-wise analysis of trajectories, success rates, and reward diversity. Unique values are shown with their corresponding total values in parentheses.

- **Unique Trajectories:** For tasks like 380, 155, and 140, the model consistently generated only a few successful trajectories, indicating that these tasks allow minimal variation in the solution space.

- **Success Rate:** Tasks such as 380, 155, and 87 achieved a 100% success rate, demonstrating that GFlowNet effectively explores correct solutions. However, for task 150, only 10% of unique trajectories were successful, suggesting a more complex solution space.

- **Reward Diversity**: Tasks with lower reward entropy (e.g., 380 and 140) reflect a solution space with limited diversity. In contrast, tasks like 87 exhibit higher entropy, indicating a greater variety of valid solutions.

## B.4 Off-Policy Training Analysis

**Experiment Setup** We explored off-policy training using a replay buffer with various sampling methods:

- **Priority Sampling (PRT)**: Prioritizes high-reward experiences.

- **Epsilon-Greedy Sampling**: Randomly samples with probability $\epsilon$ and high-reward trajectories otherwise.

- **Fixed Ratio Sampling**: Samples a fixed mix (e.g., 8:2 or 9:1) of high- and low-reward trajectories.

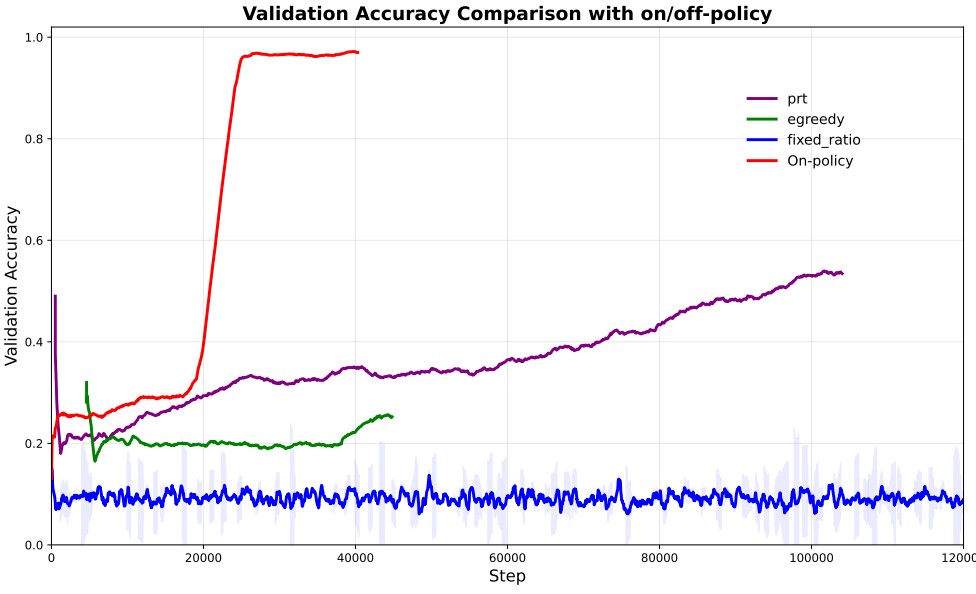

Figure 15: Validation accuracy comparison between on-policy and off-policy training methods. On-policy training converges much faster than off-policy sampling methods.

**Results and Analysis**   Off-policy training with PRT sampling achieved lower variance across seeds but required significantly more steps (approximately 120,000) to converge, compared to on-policy training which reached near-perfect accuracy within 1,000 steps. This trade-off indicates that while off-policy methods enhance stability, their slower convergence makes on-policy training more practical for ARC problems.

**(Additional Analysis)** In particular, because ARC problems are highly sparse in rewards and often solved with short sequences, quickly stumbling upon a good trajectory is crucial. Off-policy learning, which replays a mix of older experiences, can excessively re-emphasize suboptimal early trajectories unless carefully managed. On-policy methods, by contrast, adapt more rapidly to newly discovered successful paths—leading to faster improvement.

## C  Algorithm

---

**Algorithm 1** GFlowNet Architecture for ARC Task

---

**Input:** ARC environment $env$ with input $x_k$, forward policy parameters $\theta_F$, backward policy parameters $\theta_B$, total flow $Z_\theta$, reward function $r(s)$, max episode length $T$, Replay buffer BUFFER (optional)

**Output:** Trajectory $\tau$

$s \leftarrow s_0 \leftarrow x_k$

**foreach** *example* **do**

    **foreach** *training step* **do**

        $\tau \leftarrow [s]$

        $\log p_F^{total} \leftarrow 0$

        $\log p_B^{total} \leftarrow 0$

        $t \leftarrow 0$

        **while** $t < T$ **and** $\neg\, env.is\_done(s)$ **do**

            $t \leftarrow t + 1$

            $z \leftarrow \pi_F(s; \theta_F, \theta_B)$

            Split $z$ into $z_F$, $z_B$

            `# Step 1: Forward Pass`

            $p_F \leftarrow \text{Softmax}(z_F)$

            Sample $X_i \sim \text{Geometric}(p_{F_i})$ **//** $X_i = \{3, 14, 2, \dots, 7\}$

            $a^* \leftarrow \arg\min_i X_i$

            $\log p_F \leftarrow \log \text{GeomPMF}(X_{a^*})$

            $s' \leftarrow env.\text{step}(s, a^*)$

            $\tau \leftarrow \tau \cup \{s'\}$

            `# Step 2: Backward Pass`

            $p_B \leftarrow \text{Softmax}(z_B)$

            Sample $a' \sim \text{Categorical}(p_B)$

            $\log p_B \leftarrow \log p_B(a')$

            `# Step 3: Reward`

            $r(s') \leftarrow r_{final}$

            **if** *cycle is detected* **then**

                $r(s') \leftarrow r(s') - \lambda C(\tau)$

            **end**

            $\log p_F^{total} \leftarrow \log p_F^{total} + \log p_F$

            $\log p_B^{total} \leftarrow \log p_B^{total} + \log p_B$

            Store in BUFFER if off-policy (optional)

            $s \leftarrow s'$

        **end**

        $loss \leftarrow (\log Z_\theta + \log p_B^{total} - \log p_F^{total} - \log r(s))^2$

        Optimize $\theta_F$, $\theta_B$ to minimize $loss$

        Off-policy updates using BUFFER (optional)

        Update sampling model (optional)

    **end**

**end**

**return** $\tau$

---

# D   Comparison of ARC Trajectory Datasets and Generation Methods

In this section, we compare ARC trajectory datasets and generation methods according to their supervision type and augmentation strategy. We organize existing approaches into four main categories: (1) **Human-written** trajectories collected through annotation or crowdsourcing (e.g., ARCTraj (Kim et al., 2025b), ARC-Interactive (Strandgaard, 2024), H-ARC (LeGris et al., 2024)); (2) **Rule-based** trajectories derived from hardcoded logic or symbolic programs (e.g., SOLAR (Kim et al., 2024)); (3) **Model-generated** trajectories produced via generative models such as GFlowNet (Ours); and (4) **Other augmentations** such as I/O pair-based synthesis (e.g., RE-ARC (Hodel, 2024), BARC (Li et al., 2025), BabyARC (Wu, 2021)) and task/label-based formulations (e.g., LARC (Acquaviva et al., 2022), ConceptARC (Moskvichev et al., 2023)).

This categorization helps clarify the trade-offs between trajectory diversity, scalability, and generalization. In particular, our GFlowNet-based method (3) stands out by generating diverse and plausible reasoning paths directly from input-output examples, without relying on either human supervision or ground-truth programs. This makes our approach especially suitable for ARC, where solvers must generalize to unseen tasks at test time without access to task-specific demonstrations.

Additionally, unlike other methods, our method uniquely supports the generation of multiple distinct trajectories for each input grid, enabling a richer and more flexible reasoning process. To support this claim, we provide a detailed empirical and structural comparison with other methods below. Table 21 compares these datasets in terms of trajectory format, augmentation capabilities, and multi-trajectory support per grid. Table 22 then presents a quantitative comparison of trajectory scale and diversity across selected methods.

| Method | Task Count | Data Generation | Trajectory Format | Multi-Traj per Grid |
|---|---|---|---|---|
| **Ours** | 7 | ✓ | ✓ | ✓ |
| SOLAR (Kim et al., 2024) | 20 | ✓ | ✓ | ✗ (1 Traj/Grid) |
| ARCTraj (Kim et al., 2025b) | 400 | ✗ (Human Traj) | ✓ | ✓ |
| ARC-Interactive (Strandgaard, 2024) | 400 | ✗ (Human Traj) | ✓ | ✓ |
| H-ARC (LeGris et al., 2024) | 800 | ✗ (Human Traj) | ✓ | ✓ |
| RE-ARC (Hodel, 2024) | 400 | ✓ | ✗ (I/O Grid) | ✗ |
| BARC (Li et al., 2025) | 400 | ✓ | ✗ (New Task) | ✗ |
| BabyARC (Wu, 2021) | 800 | ✓ | ✗ (New Task) | ✗ |
| LARC (Acquaviva et al., 2022) | 400 | ✓ | ✗ (Task Description) | ✗ |
| ConceptARC (Moskvichev et al., 2023) | 160 | ✓ | ✗ (Task Label) | ✗ |

Table 21: Comparison of data augmentation and trajectory support across ARC-AGI methods. **Ours** is the only method that augments multiple trajectories per input grid.

Among the existing datasets, human-collected trajectories such as ARCTraj (Kim et al., 2025b), ARC-Interactive (Strandgaard, 2024), and H-ARC (LeGris et al., 2024) offer valuable insights into how people solve ARC tasks. However, these datasets typically cover a large number of tasks, but each task only has a single input grid, limiting their capacity to explore diverse solving behaviors from different inputs. Moreover, the cost of collecting multiple trajectories per input grid is prohibitively high in human studies.

In contrast, our method is the only approach that generates multiple trajectories per input grid through data augmentation. Despite using only seven target tasks, we synthesize 1,601 valid trajectories with 162 unique ones (Table 22). This results in a trajectory density of 228.7 and a trajectory diversity of 23.1, both of which

are significantly higher than those of other methods. As shown in Table 22, our trajectory diversity is even comparable to the trajectory density of human-collected datasets, which typically ranges from 19.7 to 26.7. Since diversity is upper-bounded by density, this suggests that our method achieves at least as much diversity as those human-collected datasets.

These comparisons (Tables 21 and 22) highlight the effectiveness of our augmentation method in producing a rich set of diverse solving trajectories from a compact task set, without requiring large-scale human annotation.

| Method | Valid Traj. | Unique Traj. | Target Tasks | Grids per Task | Trajectory Density | Trajectory Diversity |
|---|---|---|---|---|---|---|
| **Ours** | 1,601 | 162 | 7 | 1 | 228.7 | 23.1 |
| SOLAR (Kim et al., 2024) | 10,000 | 10,000 | 20 | 500 | 1.0 | 1.0 |
| ARCTraj (Kim et al., 2025b) | 10,672 | - | 400 | 1 | 26.7 | - |
| ARC-Interactive (Strandgaard, 2024) | 8,374 | - | 400 | 1 | 20.9 | - |
| H-ARC (LeGris et al., 2024) | 15,744 | - | 800 | 1 | 19.7 | - |

Table 22: Comparison of trajectory density and diversity across human-collected and augmented trajectories. **Trajectory Density** is calculated as the number of valid trajectories divided by the number of input grids. In contrast, **Trajectory Diversity** is the number of unique (non-duplicate) trajectories per input grid. Notably, the diversity of **Ours** (23.1) is comparable to the trajectory density of human-collected datasets such as ARCTraj, ARC-Interactive, and H-ARC. Since diversity is upper-bounded by density, this suggests that the diversity of **Ours** is at least on par with, if not superior to, those datasets.

# E   Efficiency and Scalability of GFlowNet Augmentation

**GFlowNet-based augmentation is not only effective, but also computationally efficient and structurally scalable. We report the time required to generate 100K valid trajectories for each ARC task used in our downstream experiments. As summarized in Table 23, most tasks complete this process in under an hour, with generation rates exceeding 2K trajectories per minute on average. Given that the ARC Prize evaluation protocol allows only 6 minutes per task, this level of speed implies that even *test-time augmentation* is feasible, yielding thousands of trajectories within a fraction of the allotted time.**

This level of efficiency demonstrates that our method can serve as a practical augmentation tool—even in scenarios with tight compute budgets or real-time constraints. Notably, even the slowest tasks still maintain generation rates above 1,180/min, reinforcing the method's robustness across task variability.

Table 23: Time and throughput required to generate 100,000 trajectories per task. Our framework consistently achieves multi-thousand-per-minute generation rates, enabling fast and scalable augmentation even under strict time constraints.

| Task # (ID) | Time Required (for 100k) | Generation Rate (trajectories/min) |
|---|---|---|
| 87 (3c9b0459) | 53 min | 1,880/min |
| 140 (6150a2bd) | 52 min | 1,920/min |
| 150 (67a3c6ac) | 50 min | 2,000/min |
| 155 (68b16354) | 27 min | 3,700/min |
| 179 (74dd1130) | 46 min | 2,170/min |
| 241 (9dfd6313) | 83 min | 1,200/min |
| 380 (ed36ccf7) | 84 min | 1,180/min |
| **Average** | **56.4 min** | **2,000/min** |

Beyond raw throughput, our framework also demonstrates *structural scalability* in the form of trajectory diversity. To assess how diversity grows over time, we measured the number of **unique successful trajectories** as a function of the total number of trajectories generated.

As shown in Figure 16, all tasks exhibit a consistent **logarithmic decay pattern** in the rate of discovering new unique solutions. During the early stages, new solution strategies are discovered rapidly, but the rate of discovery decreases logarithmically as the solution space becomes progressively saturated. This behavior aligns with the long-tailed structure of human reasoning, where common strategies emerge early, and rarer patterns require deeper exploration. **Notably, while all tasks follow this logarithmic trend, there are substantial variations between different problems**, with some tasks (like Problem 150) reaching saturation much faster than others (like Problems 86 and 140), reflecting the inherent complexity differences across ARC tasks.

This logarithmic scaling property has important implications. First, it allows practitioners to make informed tradeoffs between diversity and compute budget. Even small-scale generation (e.g., 10k–20k samples) captures a significant portion of the solution space, while longer runs offer diminishing but non-negligible improvements. The logarithmic nature of the decay means that the marginal benefit of additional sampling decreases predictably, enabling efficient resource allocation.

Second, this structure suggests that GFlowNet-based augmentation is well-suited for dynamic test-time settings. Depending on resource availability, one can determine how much augmentation is "enough" based on the task complexity or diversity needs. The problem-specific variations observed in Figure 16b indicate that some tasks may benefit from extended sampling while others reach effective saturation more quickly, allowing for adaptive resource allocation strategies.

Third, our framework is well-aligned with the real-time constraints of ARC-like benchmarks. For instance, the ARC Prize evaluation protocol allots approximately **6 minutes per task** to solvers, including any

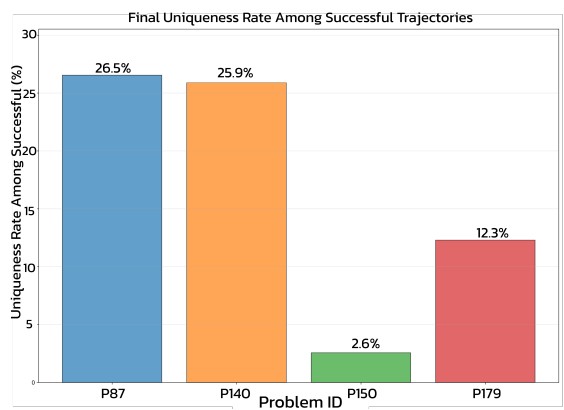
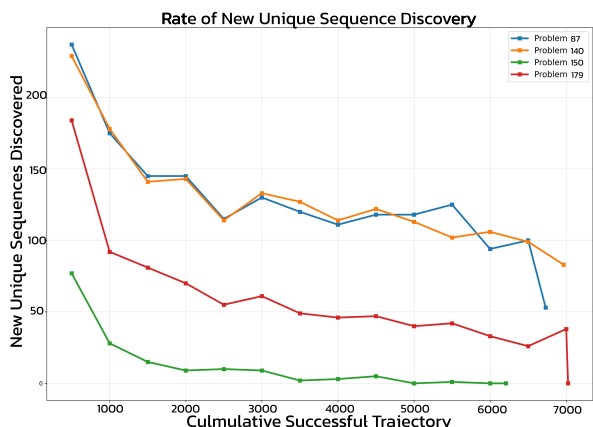

(a) Final uniqueness rates vary significantly across tasks, ranging from 2.6% to 26.5%, indicating different levels of solution diversity potential.

(b) Rate of new unique sequence discovery follows a logarithmic decay pattern, with notable variations in decay rates across different problems.

Figure 16: Trajectory diversity analysis across ARC tasks. (a) The final uniqueness rate among successful trajectories shows significant task-dependent variation. (b) The rate of discovering new unique sequences decreases logarithmically over time, with problem-specific decay characteristics reflecting varying solution space complexity.

computation performed during test time. Within this limited window, our method can generate thousands of candidate trajectories per task, thanks to its throughput of over 2,000 trajectories per minute.

This enables practical test-time augmentation, where diverse solution candidates can be synthesized and evaluated on the fly. Even allocating just 2–3 minutes of the 6-minute window for trajectory generation yields tens of thousands of diverse paths, offering valuable input to downstream symbolic solvers or neural rerankers. The logarithmic discovery pattern ensures that even short generation runs capture the most promising solution strategies, making our augmentation pipeline not only practical offline but also feasible for real-time AGI scenarios requiring rapid reasoning under constraints.

