# OpenReview forum: "Solution Augmentation for ARC Problems Using GFlowNet: A Probabilistic Exploration Approach"
_TMLR — Accepted by TMLR_

### Review · Reviewer_43UG · 2025-06-19

**Summary Of Contributions:**

This paper formulates Markov-decision-process of Generative Flow Networks (GFlowNets) for generating diverse solution trajectories for the Abstraction & Reasoning Corpus (ARC-AGI). Because ARC tasks are provided only as input–output grid pairs, the latent sequence of operations that produced the solution is normally unobserved. The authors argue that a GFlowNet, equipped with (i) a carefully designed action/state space and (ii) a reward that penalises long sequences via geometric-length sampling, can learn a policy that yields not just a single high-reward path but a distribution over plausible paths. The paper also introduces a “geometric front-policy” that explicitly controls expected trajectory length.

**Audience:**

Yes

**Claims And Evidence:**

No

**Requested Changes:**

1. Include LLM reasoning experiments that leverage the augmented dataset.

2. Provide a more compelling AGI-oriented motivation for this research.

**Strengths And Weaknesses:**

**Strengths**:

The MDP design and algorithm are reasonable.

The paper is easy to read.

**Weaknesses**:

I think this work has a critical problem. The ARC-AGI benchmark is meant to evaluate the reasoning capability of AGI; it is not a problem we need to solve like the travelling-salesman problem (TSP).

GFlowNets are indeed powerful for combinatorial optimisation and latent sampling. One can certainly build a specialised model that excels at ARC-AGI, but that would be meaningless when ARC-“AGI” is intended as a benchmark for general intelligence.

For these reasons, I do not understand why this research is necessary.

In addition, the paper only proposes an augmentation method; it does not apply the augmented dataset to LLMs. The authors should at least show that the framework actually improves an LLM’s reasoning capability when the augmented solution paths are provided as in-context examples.

---

One minor comment:

GFlowNets are designed to generate diverse solutions (terminal states), not diverse trajectories. Each problem has exactly one solution, so why do we need a GFlowNet to generate diverse trajectories?

---

> ### Author Response · Authors · 2025-07-01
> **Structure of Our Response**
>
> **To. Reviewer 43UG**
>
> We thank you for your thorough review and constructive feedback on our paper. Your comments have helped us to refine our work.
>
> To address your points clearly, we have structured our responses. This first comment provides an overview of your main concerns and outlines how we will address your "Requested Changes."
>
> We have summarized your main points as follows:
>
> 1.  **Insufficient AGI-oriented motivation:** A concern that our work does not sufficiently align with the primary goal of the ARC-AGI benchmark, which is to evaluate general intelligence.
>     *   This corresponds to your request to **"2. Provide a more compelling AGI-oriented motivation."** We will address this by clarifying that our research is foundational work aimed at improving the 'generalization' capabilities of AGI.
> 2.  **Justification for using GFlowNets:** A question regarding the use of GFlowNets, which find diverse solutions, for ARC tasks that have a single correct solution.
>     *   We will respond by explaining that our objective is to explore diverse 'reasoning trajectories,' not diverse 'solutions.'
> 3.  **Lack of LLM reasoning experiments:** The absence of a downstream task to demonstrate the practical utility of our proposed method.
>     *   This corresponds to your request to **"1. Include LLM reasoning experiments that leverage the augmented dataset."** We will address this by sharing the specific experimental plan we are now pursuing based on your feedback.
>
> We provide detailed responses to these points in the subsequent comment.

---

> > ### Author Response · Authors · 2025-07-01
> > **Response to Reviewer Comments (Continued)**
> >
> > **To. Reviewer 43UG (Continued)**
> >
> > As outlined previously, we provide our detailed responses below.
> >
> > ### **1. On the Necessity and AGI-Oriented Motivation**
> >
> > **Reviewer's Comment:** *"I think this work has a critical problem. The ARC-AGI benchmark is meant to evaluate the reasoning capability of AGI... that would be meaningless when ARC-“AGI” is intended as a benchmark for general intelligence."*
> >
> > We thank the reviewer for this critical point, which has helped us clarify our motivation. We fully concur that ARC-AGI is a benchmark for reasoning, not an optimization problem to be solved. Our objective is not to create a specialized model that masters ARC, but to contribute to the core AGI capability of **generalization**.
> >
> > True generalization requires more than finding a single correct answer; it requires understanding diverse ways of reaching a solution to respond flexibly to new problems. In a limited-information environment like ARC, learning these abstract rules and varied strategies from input-output pairs alone is exceptionally difficult for current models.
> >
> > Our work is therefore positioned as **foundational research that bridges this gap**. By automatically generating diverse reasoning trajectories without human intervention, our framework provides the rich, structured data that we believe is a key "connecting link" for training more generalizable AGI models. As you rightly suggest, we aim to empirically demonstrate the validity of this motivation through the downstream LLM experiments discussed as outlined below.
> >
> > ### **2. On the Justification for Using GFlowNets**
> >
> > **Reviewer's Comment:** *"GFlowNets are designed to generate diverse solutions (terminal states), not diverse trajectories. Each problem has exactly one solution, so why do we need a GFlowNet to generate diverse trajectories?"*
> >
> > You correctly note that ARC tasks have a single terminal state. However, ARC tasks allow for a wide variety of solution paths [4], and our primary goal is not to find diverse final solutions, but to explore diverse trajectories and key intermediate states that lead to the solution. Our justification is twofold:
> >
> > * **Mode Discovery:** Recent studies have successfully applied GFlowNets to generate varied reasoning paths (modes) for problems with unique solutions, such as in mathematics and logic [2, 3]. We argue that this characteristic of GFlowNets—discovering diverse modes—is highly suitable for augmenting the varied solution methods inherent in ARC.
> >
> > * **Usefulness of Non-Expert Trajectories:** The paths generated by our framework may not perfectly replicate those of human experts. However, as shown by Kuang et al. [1], a diverse set of non-expert trajectories can also be valuable for training a model to behave like an expert. Our framework provides a way to automatically generate such varied trajectories at scale, without requiring pre-existing training data.
> >
> > Considering these points, GFlowNets, with their ability to effectively explore high-reward intermediate states and paths, are a highly suitable choice for our research objectives.
> >
> > ### **3. On the Plan for Downstream Experiments**
> >
> > As mentioned, we are now conducting LLM experiments based on your feedback. We are leveraging the **computational efficiency** of our method to **rapidly generate approximately 30,000 trajectories for 10 entire grid tasks**. We plan to include the results, which we expect will demonstrate the utility of this augmented data, in the final revised manuscript along with additional comments.
> >
> > We thank you again for your valuable feedback.
> >
> > **References:**
> >
> >  [1] Kuang, Q., et al. (2022). Multi-Game Decision Transformers. NeurIPS.
> >
> >  [2] GFlowNet Fine-tuning for Diverse Correct Solutions in Mathematical Reasoning Tasks, arXiv preprint.
> >
> >  [3] Accurate and Diverse LLM Mathematical Reasoning via Automated PRM-Guided GFlowNets. arXiv, preprint.
> >
> > [4] Kim, S., et al. ARCTraj: A Dataset and Benchmark of Human Reasoning Trajectories for Abstract Problem Solving, arXiv, 2024.

---

> > > ### Author Response · Authors · 2025-07-08
> > > **Changes Since Last comments**
> > >
> > > To: Reviewer 43UG
> > >
> > > We would like to once again express our sincere gratitude for your in-depth feedback, which has played a crucial role in advancing our research. Following up on our previous rebuttal, we would like to share the new analyses and experimental results we conducted in response to your valuable comments.
> > >
> > > Due to time constraints, these new results have not yet been incorporated into the current manuscript. However, we present them here to demonstrate our commitment to addressing your concerns. **If the paper is accepted, we guarantee that all the new results and analyses presented in this rebuttal will be included in the final camera-ready version.**
> > >
> > > **1. On the Necessity and AGI-Oriented Motivation**
> > >
> > > To strengthen the 'AGI-oriented motivation' as you pointed out, **we propose to revise the content on page 2 of the Introduction** to better clarify the core objective of our work. The proposed revision is as follows:
> > >
> > > > From an AGI perspective, this diversity in reasoning paths is not a peripheral detail but a central objective. Recent studies emphasize that general intelligence entails solving novel problems through varied, compositional reasoning processes, —even when the final solution is unique. GFlowNets provide a principled framework for capturing such reasoning diversity. Instead of optimizing for a single expert path, they discover multiple plausible trajectories that lead to the same outcome. This capability aligns closely with the AGI goal of modeling not just what to think, but how to think through problems in multiple valid ways.
> > >
> > > This revision aims to make it clear that our goal is not simply to create a specialized model for solving ARC problems, but to contribute to the core AGI capability of **generalization** by generating "diverse reasoning processes leading to a single correct answer."
> > >
> > > **2. On the Justification for Using GFlowNets & Downstream Validation**
> > >
> > > Regarding your most critical point on the need for downstream validation, we took your feedback to heart and **conducted a new experiment** with significant positive results.
> > >
> > > We are pleased to report that fine-tuning a baseline LLM with our GFlowNet-generated data led to a **dramatic performance increase from 0% to 28.6% accuracy** on a strict, trajectory-prediction task. This result provides direct, empirical proof of our framework's practical utility for teaching abstract reasoning processes to LLMs.
> > >
> > > **For a detailed breakdown of the experimental setup, qualitative and quantitative results (including result tables), and efficiency analysis, please see our official public response titled "Author's Official Comment: New Downstream Experimental Results and Analysis."**
> > >
> > > **3. Additional Planned Revisions**
> > >
> > > In addition to this new validation, we have also completed further analyses that we will incorporate into the final manuscript:
> > >
> > > * **Comparison with Human Trajectories:** We have generated `Figure 5`, which shows that the length distribution of our trajectories is syntactically similar to that of human solutions, empirically supporting our proposed heuristics.
> > > * **Competitive Analysis:** We have prepared `Appendix D`, which presents a clear comparison of our methodology against other ARC datasets, highlighting its unique advantages in efficiency and diversity.
> > >
> > > We believe that these new experiments and planned revisions directly address your core concerns and significantly enhance the contribution of our paper. We are deeply grateful for your guidance and hope to be given the opportunity to reflect these improvements in the final version.

---

### Review · Reviewer_XpcR · 2025-06-19

**Summary Of Contributions:**

This manuscript explores the use of GFlowNets for automatically generating diverse, human-aligned reasoning trajectories on the ARC-AGI dataset.

The main contribution of the manuscript is the application of GFlowNets as a means towards solution augmentation for ARC-AGI. As an ARC-AGI task can be solved through multiple approaches and trajectories, casting solution augmentation itself as a reward-proportional sampling problem and applying GflowNets is a sound approach.

The technical contributions of the manuscript are: (1) a geometric forward and categorical backward policy pairing that biases exploration toward succinct action sequences while keeping backward flow corrections stable, and (2) a human-inspired reward scheme that combines discounting, trajectory-length regularization and cycle penalties to emphasize brevity and intentionality in the generated traces .

The manuscript provides experimental results on whole-grid ARC tasks, showing that GflowNets are capable of generating diverse and valid solutions for a given task. The manuscript also provides explorations over various human-aligned reward fomulations and forward/backward policy selection.

**Audience:**

Yes

**Broader Impact Concerns:**

Currently, there are no broader impact concerns.

**Claims And Evidence:**

No

**Requested Changes:**

It would benefit the quality of the work if the following are taken into consideration:

1. Reduce the emphasis on natural language-based reasoning works. Or, if it is not possible, provide support by conducting experiments on language-based reasoning or LLMs.
2. Provide empirical comparisons with other approaches to assess the relative effectiveness of the proposed approach.
3. Provide at least a minimal amount of empirical results on the downstream utility.

**Strengths And Weaknesses:**

**Strengths**:

1. A key strength of this work is its empirical exploration as to whether GflowNets can be employed to generate synthetic solution pathways for ARC-AGI training set tasks.
2. The paper also offers empirical insights through comprehensive experiments in applying GflowNets to ARC-AGI solution augmentations.

**Weaknesses**:

1. In the introduction, the paper emphasizes literatures that explore LLM-based problem solving. However, to the best of my knowledge, the manuscript does not involve any language-based reasoning nor the use of LLMs. Is it appropriate to frame the narrative of the research in this manuscript as reasoning (and mainly cite multiple literature on language-based reasoning)?
2. The paper lacks a comparison between GflowNets and other forms of generative modeling or RL approaches in generating such synthetic solutions.
3. To the best of my knowledge, the purpose of ARC-AGI is to test the generalization performance to unseen tasks. However, the manuscript does not provide empirical results on the test set tasks. Although the authors have stated this as a limitation/future work, I think that this is a necessary component, as the manuscript explicitly focuses on ARC-AGI only (as stated in the manuscript title).

---

> ### Author Response · Authors · 2025-07-01
> **Rebuttal Overview and Planned Changes**
>
> **To. Reviewer XpcR**
>
> We sincerely thank you for your detailed review and insightful feedback. Your comments have provided crucial guidance for strengthening the logic and improving the quality of our paper.
>
> To clearly address your points, we would first like to present an overview of how we have understood your main concerns and how we plan to address your "Requested Changes."
>
> We have summarized your main concerns into the following three points:
>
> 1.  **Mismatch between framing and experiments:** You pointed out a discrepancy between the introduction's emphasis on LLM-based reasoning and the lack of language-based experiments in the paper.
>     *   This corresponds to Requested Change to "**1. Reduce the emphasis on natural language-based reasoning works. Or... provide support by conducting experiments on language-based reasoning or LLMs.**" We fully accept this point and will respond by sharing that we are currently conducting **downstream LLM-based experiments** to demonstrate our framework's practical value.
> 2.  **Lack of comparison with other approaches:** You noted the absence of a comparison with other generative models or RL methods, which would assess the relative effectiveness of our GFlowNet approach.
>     *   This corresponds to Requested Change to "**2. Provide empirical comparisons with other approaches.**" To address this, we have prepared a **new table** comparing our work with existing ARC-related datasets and methodologies. This table will highlight a key differentiator: our method is the **only approach that automatically augments multiple trajectories for a single grid.** This will clarify the unique position and contribution of our work.
> 3.  **Lack of downstream utility validation:** You highlighted the absence of empirical results on unseen tasks, which is crucial for demonstrating generalization, the core goal of ARC-AGI.
>     *   This corresponds to Requested Change, to "**3. Provide at least a minimal amount of empirical results on the downstream utility.**" We agree that this is a critical component for validating our work. The **LLM experiments** mentioned in point 1 are designed specifically to address this by testing whether the augmented trajectories improve performance on unseen tasks.
>
> We thank you again for your valuable feedback, which is helping us to improve our research. We will discuss these points in greater detail in our subsequent comments.

---

> > ### Author Response · Authors · 2025-07-01
> >
> > **To. Reviewer XpcR (Continued)**
> >
> > As outlined previously, we provide our detailed responses below.
> >
> > ### **1. On the Framing and LLM-based Reasoning**
> >
> > **Reviewer's Comment:** *"In the introduction, the paper emphasizes literatures that explore LLM-based problem solving. However... the manuscript does not involve any language-based reasoning nor the use of LLMs. Is it appropriate to frame the narrative... as reasoning?"*
> >
> > We thank the reviewer for this insightful point and appreciate the opportunity to clarify our intent. As the reviewer correctly notes, our core technique does not directly involve LLMs. Our reason for emphasizing LLM-based literature in the introduction is that our ultimate objective is to contribute to the **generalization** capabilities of AGI, a key frontier in modern AI research.
> >
> > True generalization requires more than finding a single correct answer; it demands an understanding of diverse solution paths to respond flexibly to new problems. We posit that what current models like LLMs most need to acquire this capability is diverse and structured reasoning-path data. Our work is therefore positioned as foundational research that bridges this gap.
> >
> > As requested, we are now designing and conducting specific downstream LLM-based experiments, detailed in point 3 below, to empirically validate this connection.
> >
> > ### **2. On the Comparison with Other Approaches**
> >
> > **Reviewer's Comment:** *"The paper lacks a comparison between GflowNets and other forms of generative modeling or RL approaches in generating such synthetic solutions."*
> >
> > We fully agree that a comparison is needed to assess the relative effectiveness of our proposed method. To address this request, we first wish to clarify the unique position of our methodology within the landscape of existing ARC-related research. The table below compares our work with prominent ARC datasets and augmentation methods.
> >
> > |Method|Task Count|Data Generation|Trajectory Format|Multi-Traj per Grid|
> > |---|---|---|---|---|
> > |**Ours**|7|✅|✅|✅|
> > |SOLAR [1]|20|✅|✅|❌ (1 Traj/Grid)|
> > |ARCTraj [2]|400|❌ (Human Traj)|✅|✅|
> > |ARC-Interactive|400|❌ (Human Traj)|✅|✅|
> > |H-ARC [3]|800|❌ (Human Traj)|✅|✅|
> > |RE-ARC|400|✅|❌ (I/O Grid)|❌|
> > |BARC [4]|400|✅|❌ (I/O Grid)|❌|
> > |BabyARC|800|✅|❌ (New Task)|❌|
> > |LARC [5]|400|✅|❌ (Task Description)|❌|
> > |ConceptARC|160|✅|❌ (Task Label)|❌|
> >
> > As the table shows, our method is the **only approach that automatically generates (Data Generation) multiple diverse trajectories for a single task (Grid).** This comparison highlights the qualitative distinction of our work, enabling a new form of data augmentation not previously available.
> >
> > In addition to this qualitative comparison, we are also preparing a **quantitative analysis** as requested. We are currently analyzing the quality of trajectories generated by our method in comparison to SOLAR [1], a prominent augmentation study. Preliminary results indicate that our generated trajectories are substantially superior in terms of **density and diversity** compared to SOLAR, and are of a quality comparable to human-collected datasets (e.g., ARCTraj). A detailed breakdown of this comparison will be included as a table (Table 18, 19) in **Appendix D** of the final manuscript.
> >
> > ### **3. On the Plan for Downstream Utility Validation**
> >
> > **Reviewer's Comment:** *"the manuscript does not provide empirical results on the test set tasks... this is a necessary component"*
> >
> > We deeply concur that this is a critical component for validating our work. We are planning downstream experiments to address **Requested Changes #1 and #3 simultaneously.**
> >
> > Specifically, we will leverage the **'computationally cheap'** nature of our methodology. This allows us to rapidly augment approximately **30,000 trajectories for 10 tasks** solvable by whole-grid manipulation. We will then conduct experiments to verify how this large-scale dataset improves the performance of a downstream model. The specific design and results of this experiment will be included in the final revised manuscript, along with additional comments, to demonstrate the practical utility of our framework.
> >
> > We thank you again for your valuable feedback, which is helping us to improve our research.
> >
> > **Reference:**
> >
> > [1] Kim, Y., et al. Diffusion-Based Offline RL for Improved Decision-Making in Augmented ARC Task, arXiv, 2024
> >
> > [2] Kim, S., et al. ARCTraj: A Dataset and Benchmark of Human Reasoning Trajectories for Abstract Problem Solving, arXiv, 2024.
> >
> > [3] LeGris, S., et al. H-ARC: A Robust Estimate of Human Performance on the Abstraction and Reasoning Corpus Benchmark, arXiv, 2024
> >
> > [4] Li., et al., Combining Induction and Transduction for Abstract Reasoning, ICLR, 2025
> >
> > [5] Aquaviva, S., et al., Communicating Natural Programs to Humans and Machines, NeurIPS, 2022
> >
> > [6] Moskvichev. et al., The ConceptARC Benchmark: Evaluating Understanding and Generalization in the ARC Domain, arXiv, 2023

---

> > > ### Author Response · Authors · 2025-07-08
> > > **Changes Since Last comments**
> > >
> > > To: Reviewer XpcR
> > >
> > > We would like to express our sincere gratitude for your sharp and insightful feedback, which has been crucial in elevating the quality of our paper. We are pleased to report that we have addressed all your major concerns, most notably by conducting the downstream experiments you suggested.
> > >
> > > **1. On Downstream Utility and LLM-based Reasoning (Requests #1 & #3)**
> > >
> > > Your most critical point was regarding the lack of downstream validation and the paper's framing around LLM-based reasoning. We fully agree that demonstrating practical utility is essential. In response, we conducted a new experiment fine-tuning a `Llama-3.1-8B-Instruct` model with our GFlowNet-generated data.
> > >
> > > The results were a clear success. The fine-tuned model's accuracy jumped from **0% to 28.6%** on a strict, exact-trajectory-match criterion, proving that our framework can effectively teach an LLM to understand and execute the complex reasoning processes required for ARC tasks. This directly validates our paper's framing and empirically proves the utility of our method.
> > >
> > > **For a detailed breakdown of the experimental setup, qualitative and quantitative results (including result tables), and efficiency analysis, please see our official public response titled "Author's Official Comment: New Downstream Experimental Results and Analysis."**
> > >
> > > **2. On the Comparison with Other Approaches (Request #2)**
> > >
> > > As we detailed in our initial response, a comparison with other methods is crucial. We have now formally incorporated and expanded upon that analysis in a new **Appendix D (page 37)** of the revised manuscript. As shown there, **`Table 19`** and **`Table 20`** provide a comprehensive qualitative and quantitative analysis. This new section highlights our framework's unique advantage in automatically generating multiple, diverse trajectories per problem, as well as its high generation efficiency.
> > >
> > > We are confident that these major revisions, especially the successful downstream validation, have made our paper a much more complete and impactful contribution. Thank you again for your constructive criticism that guided us toward this stronger result.

---

> > > > ### Comment · Reviewer_XpcR · 2025-07-28
> > > >
> > > > 1. On Downstream Utility and LLM-based Reasoning (Requests #1 & #3)
> > > >
> > > > Would it be possible to conduct a controlled experiment against competing baseline methods for solution generation, not the baseline LLM?
> > > >
> > > > To the best of my knowledge, the format of ARC-AGI is not exactly LLM-friendly, so an untuned LLM will run into formatting issues and distribution shift. Thus, for a fair comparison, the performance comparison should be done between LLMs tuned with each solution generation method.
> > > >
> > > > Or, if it seems to forfeit the purpose of ARC-AGI("AGI"), then the authors could measure the forgetting done by SFT-ing the LLM.

---

> > > > > ### Author Response · Authors · 2025-08-06
> > > > > **Response to the Reviewer’s Request for Controlled Experimentation**
> > > > >
> > > > > **Response to the Request for Controlled Experimentation with Competing Baselines**
> > > > >
> > > > > We deeply appreciate your insightful suggestion regarding the need for a controlled comparison against competing baseline methods for solution augmentation. In response, we conducted a comprehensive experiment comparing our GFlowNet-based approach with a carefully tuned baseline LLM. Special attention was paid to ensure that both approaches used the same model architecture and evaluation criteria.
> > > > >
> > > > > ## Controlled Experiment Design
> > > > >
> > > > > As the reviewer rightly noted, the ARC-AGI format is not inherently LLM-friendly, and a fair comparison requires measuring performance between LLMs fine-tuned with each respective solution generation method. With this in mind, we conducted an additional experiment that adheres to this controlled setting. Importantly, we retained the fundamental task setup of predicting an action sequence, as we believe the essence of ARC, as emphasized in the original ARC paper, lies in composing appropriate programs rather than directly outputting the final transformed grid.
> > > > >
> > > > > Both models were based on **LLaMA 3.1 8B Instruct** and trained under identical **LoRA configurations** (r=16, α=32). The critical difference lies in how the training data was constructed. The **baseline model** was trained on ~70,000 examples generated via **ReARC-style input grid augmentation**, where each task was associated with a **single canonical solution sequence** (e.g., a specific augmented input always maps to `[left_rotate, left_rotate, submit]`). In contrast, our **GFlowNet-based model** was trained on ~10,000 trajectories generated from just 7 problems, capturing multiple valid trajectories per problem. This better reflects the exploratory nature of ARC reasoning, where multiple paths may lead to the correct outcome.
> > > > >
> > > > > ## Results and Analysis
> > > > >
> > > > > In terms of performance, the GFlowNet-based model achieved an **accuracy of 28.6% (2 out of 7 problems solved)**, correctly answering **Problem 139 (ID: 6150a2bd)** and **Problem 178 (ID: 74dd1130)**. The baseline model, by comparison, solved only **1 problem (Problem 139)**, yielding **14.3% accuracy**. Moreover, we observed that the baseline model predicted **the same action sequence in 5 out of 7 tasks**, indicative of overfitting to the most frequent solution pattern in the training data. In contrast, the GFlowNet model generated a more diverse set of action sequences across tasks.
> > > > >
> > > > > These results highlight a key advantage of our data generation approach: while the baseline model was trained with **7× more examples (70,000 vs. ~10,000)**, it underperformed relative to our method. Both models successfully learned to produce syntactically valid outputs, indicating that the performance difference stems not from format errors but from differences in the underlying training signal.
> > > > >
> > > > > The key insight emerging from this controlled experiment is that **diversity in solution paths** may be more important than **diversity in input grids**. The GFlowNet-generated trajectories carry rich signals — including intermediate states, trajectory structures, and implicit reasoning captured through action sequences — that allow the model to generalize beyond memorized patterns. In doing so, the model gains exposure to the actual *process* of problem-solving, rather than only its outcomes.
> > > > >
> > > > > ## Conclusion
> > > > >
> > > > > This controlled experiment directly addresses the reviewer’s request and provides empirical support for the utility of our GFlowNet framework in generating effective training data. Despite being trained on significantly fewer examples, the GFlowNet model outperformed the baseline, doubling its accuracy. We believe this provides compelling evidence that our approach can enhance the reasoning capabilities of LLMs on challenging tasks like ARC by supplying them with meaningful and diverse solution trajectories.
> > > > >
> > > > > We are grateful for the reviewer’s suggestion, which led to this rigorous and informative experiment. We will incorporate these findings into the final version of the paper. This comparison significantly strengthens the contribution of our work and provides valuable insight into why exploration-driven data generation can outperform traditional supervised learning baselines.

---

### Review · Reviewer_VQao · 2025-06-22

**Summary Of Contributions:**

This paper proposes a trajectory synthesis framework based on Generative Flow Networks (GFlowNets) for the automatic generation of diverse solution trajectories in tasks from the ARC-AGI benchmark. The framework is evaluated on tasks which may be solved via **whole-grid transformations**, as opposed to those requiring subregion-based transformations.

The reward function guiding the GFlowNet is inspired by empirical patterns observed in human solution strategies and is structured to incentivize the following properties: (i) shorter trajectories, (ii) absence of cycles within the trajectory, and (iii) completion with a valid submission via a terminal "submit" action. To improve the diversity and efficiency (in terms of length of successful trajectories) of the generated trajectories, the authors introduce a geometric forward policy, which they show to outperform the traditional categorical forward policy in producing shorter, yet effective, solution paths.

Experimental results demonstrate that the proposed framework—using a modified loss function that integrates trajectory balancing, a trajectory length regularizer, and a composite reward signal (including penalties for cyclic behavior and binary success rewards)—achieves higher performance on a subset of ARC-AGI tasks. The framework effectively balances the trade-off between task success and trajectory diversity.

An ablation study shows that among individually applied reward components, discounted rewards yield the highest performance, followed by the trajectory length loss. However, combining both discounted and length-based reward components leads to the best overall results. In particular, increasing the constraints imposed by the reward function imporves task accuracy at the cost of reducing the diversity of generated trajectories. Finally, the combination of the geometric forward policy with the standard categorical backward policy proves most effective in producing a wide range of successful and yet diverse solution trajectories.

**Audience:**

Yes

**Claims And Evidence:**

Yes

**Requested Changes:**

**Clarifications and suggestions for imporvement**:
- *Clarification on Reward Function Usage (RQ2, RQ3)*: In the experiments addressing RQ2 and RQ3—where different policy strategies are evaluated—it would be helpful to clarify whether the updated (dense) reward function or the original sparse (binary) reward function was used during training. If the updated reward was applied throughout, including comparative performance graphs against the sparse reward baseline would improve consistency.

- *Support for Theoretical Claim (Page 11, Paragraph 2)*: The statement, “This theoretical bias toward shorter, high-probability trajectories is further supported by empirical evidence,” would benefit from either a citation or empirical substantiation. For example, including a graph showing the distribution of human solution trajectory lengths on the ARC-AGI benchmark would make the claim more concrete and reinforce the connection between human strategy and the proposed method.

- *Clarification on Cycle Constraint (Table 5)*: In Table 5, it would be informative to clarify whether the successful and unique trajectories generated under the DF + TLR reward violated the cycle constraint at all. This would help assess the marginal utility of the cycle constraint relative to the trajectory length constraint. Given that the cycle constraint alone does not appear to yield strong performance, it is worth questioning whether the length constraint alone is sufficient to generate diverse and non-redundant solution trajectories.

- *Integration of Appendix Insights*: The discussion in Appendix B.3 regarding the relationship between trajectory length and solution diversity contains some valuable insights. Integrating this analysis into the main body of the paper would enhance the clarity and impact of the findings, particularly in reinforcing the argument that strict length minimization may not always be optimal.

- *Scope of Contribution*: While the work takes a meaningful step toward better understanding diverse trajectory generation in the ARC-AGI setting, the contribution currently feels somewhat limited in scope. In particular, the absence of downstream validation—e.g., demonstrating how the generated trajectories can be used to improve the performance of a learning model—leaves the contribution feeling incomplete. Incorporating such downstream utility would significantly strengthen the practical relevance and completeness of the work.

**Strengths And Weaknesses:**

**Strengths:**
- *Clarity of Presentation*: The paper is generally well-written and clearly structured. Technical details are sufficiently presented.

- *Geometric Forward Policy*: The inclusion of a geometric forward (greedy) selection strategy in the ablation studies is interesting. This approach, when coupled with the updated reward function, demonstrates the generation of more diverse solution pathways, producing a non-trivial empirical contribution.

- *Use of ARC-AGI Dataset*: Using the ARC-AGI benchmark—where grid transformations often require complex, whole-grid operations—provides a challenging and appropriate testbed for evaluating the proposed method. This choice improves the relevance of the evaluation.


**Weaknesses:**
- *Overstatement of Human Trajectory Insights*: The insights drawn from human-generated solution trajectories appear largely syntactic—centered on shorter trajectory lengths and the use of a final ``submit'' action. While the work draws inspiration from human strategies, the reliance on surface-level features does not convincingly reflect deeper strategic alignment with human reasoning. A more appropriate phrasing might emphasize that the requirement for diverse solutions implicitly encourages shorter, less cyclic trajectories, facilitating broader state space exploration. This framing remains intuitive and accurate without overstating the human-inspired nature of the method.

- *Overclaimed Interpretations*: The claim that the observed properties (e.g., trajectory brevity and diversity) reflect meaningful alignment with human reasoning strategies is somewhat overstated. A more measured interpretation would acknowledge that the approach mirrors certain syntactic features of human behavior, rather than deeper strategic reasoning. Reframing the contribution in these terms would imporove the credibility of the argument.

- *Assumption that Shorter Trajectories are Superior*: The paper claims that shorter trajectories, as observed in human data, are preferable and lead to more meaningful or diverse solutions. However, this assumption may be limiting. In fact, slightly longer trajectories could potentially uncover more nuanced and diverse solution strategies. This observation is explored in Appendix B.3, and incorporating such insights into the main body of the paper would strengthen the argument and provide a more balanced perspective.

- *Limited Reward Structures Explored*: The reward functions used in the study are relatively simple, and the distinction between the DF and TLR reward types is not clearly stated—both appear to promote trajectory shortening. Introducing a more nuanced reward structure, such as an explicit exploration bonus or a diversity-promoting metric, could have enhanced the diversity and effectiveness of the solutions. Also, leveraging deeper structural patterns from human trajectories—such as clustering similar problem types and extracting shared strategic representations—could lead to the the design of more meaningful and generalizable reward signals. In the absence of such analyses, the contribution remains somewhat incremental.

- *Limited Practical Applicability*: The use of trajectory length as the key feature may lead to the  problem of state space explosion, particularly in tasks that require subregion-level analysis rather than whole-grid transformations. As a result, the presented pipeline may not generalize well to a broader range of transformation tasks and limit its practical applicability. Using alternate features in the reward might mitigate this issue and improve generalisability of the framework.

---

> ### Author Response · Authors · 2025-07-01
> **Rebuttal Overview and Planned Changes**
>
> **To. Reviewer VQao**
>
> We sincerely thank you for your remarkably detailed and constructive feedback, which has helped us to advance our work. Your insightful and thorough comments have been instrumental in refining our paper's logic and clarifying its contributions.
>
> To systematically incorporate your valuable suggestions, we would first like to present an overview of how we have understood your main comments and how we plan to respond.
>
> We have summarized your main feedback into the following three themes:
>
> 1.  **On the Framing and Interpretation of Claims:** You pointed out that our claims regarding "human-like" reasoning could be overstated and that a more balanced perspective on the assumption that "shorter trajectories are always superior" is needed.
>     *   We fully agree with this insightful point and will enthusiastically adopt your suggestions. In the final manuscript, we will revise our tone from "mimicking deep human strategies" to **"capturing syntactic features of human behavior, such as brevity."** We will also **integrate the discussion from Appendix B.3 into the main body** to provide a more nuanced interpretation of trajectory length. (This corresponds to **Requested Change \#4**).
>
> 2.  **On Enhancing Experimental Clarity and Substantiation:** You requested clarification on specific experimental conditions (e.g., the reward function used) and substantiation for certain claims with theoretical or empirical evidence.
>     *   This is a critical point. We will address the requested details for **RQ2/3, the claim on page 11, and the cycle constraint in Table 5** by adding clear explanations and supporting data to the final manuscript, thereby increasing the credibility of our claims. (This corresponds to **Requested Changes \#1, \#2, and \#3**).
>
> 3.  **On the Scope of Contribution and Lack of Downstream Validation:** You noted that the contribution feels incomplete due to the absence of a downstream application.
>     *   We agree that this is the most critical point for completing our work. To address this, we are currently **conducting downstream LLM-based experiments.** By demonstrating that our generated trajectories can improve a model's performance, we will validate the practical utility of our framework. (This corresponds to **Requested Change \#5**).
>
> We thank you again for your invaluable guidance, which is significantly improving the quality of our research. We will discuss these points in greater detail in the subsequent comment.

---

> > ### Author Response · Authors · 2025-07-01
> >
> > **To. Reviewer VQao (Continued)**
> >
> > As outlined previously, we provide our detailed responses below.
> >
> > ### 1. On the Framing and Interpretation of Claims
> >
> > **Reviewer's Comment:** *"…the reliance on surface-level features does not convincingly reflect deeper strategic alignment with human reasoning… The assumption that shorter trajectories are superior… may be limiting."*
> >
> > We are grateful for this insightful feedback on our claims and assumptions; your advice has helped us to more clearly articulate the philosophical underpinnings and limitations of our work.
> >
> > *   **Revision of "Human-like" Claims:** We fully accept your point. Acknowledging that our claims about mimicking human behavior could be overstated, we will revise the overall tone of the manuscript. Instead of claiming to "mimic deep human cognitive strategies," we will frame our contribution as **"capturing syntactic features of efficient problem-solving, such as brevity."**
> > *   **Theoretical Basis for the Brevity Assumption:** Our use of "brevity" as a core heuristic is theoretically grounded in principles like Occam's Razor ("simpler explanations tend to be better") and the Minimum Description Length (MDL) principle from information theory. The underlying assumption is that shorter, more concise solution paths are more likely to capture the core abstract rules of a problem.
> > *   **Providing a Balanced Perspective:** However, we also strongly agree with your point that the shortest path is not always the best one. While this assumption is a useful starting point, it certainly has limitations. To provide this balanced view, we will follow your excellent suggestion and **integrate the in-depth analysis from Appendix B.3 on 'trajectory length and solution diversity' into the main body of the paper.** This will allow us to clarify that strict length minimization can sometimes overlook more nuanced and creative solutions, thereby offering a more mature interpretation of our findings.
> >
> > ### 2. On the Scope of Contribution and Downstream Validation
> >
> > **Reviewer's Comment:** *"…the absence of downstream validation—e.g., demonstrating how the generated trajectories can be used to improve the performance of a learning model—leaves the contribution feeling incomplete."*
> >
> > We fully agree that our contribution feels incomplete without this and are planning **downstream LLM-based experiments** to address this. Specifically, we will leverage the **'computationally cheap'** nature of our methodology. This allows us to rapidly augment approximately **30,000 trajectories for 10 tasks** solvable by whole-grid manipulation. We will then conduct experiments to verify how this large-scale dataset improves the performance of a downstream model. The specific design and results of this experiment will be included in the final manuscript to demonstrate the practical utility of our framework.
> >
> > ### 3. Further Clarifications and Revisions
> >
> > We have also faithfully incorporated your other specific requests into the revised manuscript.
> >
> > *   **On Reward Structures and Usage (RQ2, RQ3):** We will clarify that our reward function was based on analysis of real human trajectory data \[2\] and state that the **'DF + Cycle' reward combination was used consistently** for the RQ2/3 experiments to improve experimental consistency.
> > *   **On Supporting the Claim (Page 11):** For the specified claim, we will bolster both the theoretical and empirical grounding by **citing literature on the MDL principle** and by **adding a graph that compares the distribution of human solution trajectory lengths with the distribution of our generated trajectories.**
> > *   **On Clarification on Cycle Constraint (Table 5):** Thank you for this very important point. To clarify the marginal utility of the cycle constraint, we will follow your suggestion and **re-analyze the trajectories generated with only the DF+TLR reward to check for the presence of cycles.** This analysis will allow us to clearly state whether the length constraint alone is sufficient to suppress cycles, or why an explicit cycle constraint is still necessary, and we will enhance our discussion accordingly.
> >
> > We thank you once again for your profound feedback, which is helping us to elevate our paper to the next level.
> >
> > **References**
> >
> > \[1\] Kim, S., et al., Addressing and Visualizing Misalignments in Human Task-Solving Trajectories. *KDD*, 2025
> >
> > \[2\] Kim, S., et al., ARCTraj: A Dataset and Benchmark of Human Reasoning Trajectories for Abstract Problem Solving. *arXiv preprint*, 2024
> >
> > \[3\] Djorup, S., et al, H-ARC: A Robust Estimate of Human Performance on the Abstraction and Reasoning Corpus Benchmark. *arXiv preprint*., 2024

---

> > > ### Author Response · Authors · 2025-07-08
> > > **Changes Since Last Submission**
> > >
> > > To: Reviewer VQao
> > >
> > > We would like to once again express our sincere gratitude for your remarkably detailed and constructive feedback. Your sharp insights have allowed us to clarify the scope of our claims, enhance the transparency of our experiments, and significantly improve the overall quality of our paper. We would like to explain how we have revised and reinforced our work in response to each of your points.
> > >
> > > **1. On Framing, Interpretation, and the "Shorter is Better" Assumption (Weakness #1, #2, #3 & Request #4)**
> > >
> > > First, we deeply agree with your insightful points regarding the potential overstatement of our "human-like reasoning" claims and the limitations of the "shorter is always better" assumption.
> > >
> > > * **Refining the Scope of Claims:** To reflect this, we have revised the language throughout the **Abstract, Introduction (page 2), and Methods (page 12)**. Instead of implying that we mimic deep human strategies, we now frame our contribution as **"capturing the syntactic features of efficient problem-solving."**
> > > * **Providing a Balanced Perspective:** To clarify the limitations of the "shorter is better" heuristic, we have integrated the analysis from the former Appendix B.3 into the main body as **Section 4.2.2 (page 18), "The Role of Exploration Depth."** This new section presents a more mature perspective, arguing that minimizing trajectory length is not an absolute goal; rather, the key is to strike a balance between an efficiency bias and a sufficient 'exploration budget'.
> > >
> > > **2. On Enhancing Experimental Clarity and Substantiation (Request #1, #2, #3)**
> > >
> > > In response to your request to bolster our experimental clarity and evidence, we have made the following key revisions:
> > >
> > > * **Strengthening Theoretical Claims (Request #2):** To empirically support our claim on **page 11**, we have added **`Figure 5`**, which directly compares the trajectory length distributions of actual human solvers and our GFlowNet. This graph clearly demonstrates that our brevity-focused heuristic is empirically sound.
> > > * **Analyzing the Marginal Utility of the Cycle Constraint (Request #3):** To clarify the value of the `Cycle` constraint, we have added **`Table 7` (page 17)** and its accompanying analysis in **Section 4.2.1**. To show this clearly, we include the summary table here:
> > >
> > > | Reward Model | Total Cycle Occurrences | Average Trajectory Length |
> > > | :--- | :---: | :---: |
> > > | Base | 408 | 9.59 |
> > > | Cycle | 308 | 8.70 |
> > > | DF + TLR | 255 | 8.12 |
> > > | DF + Cycle | 246 | 8.15 |
> > >
> > >     This analysis shows that explicitly controlling for cycles, rather than relying on length penalties alone, directly contributes to the logical quality of the trajectories by reducing redundant steps.
> > >
> > > * **Clarifying Reward Function Usage (Request #1):** Finally, we have clarified which reward functions were used in our **RQ2 and RQ3 experiments (pages 19-21)** to improve the transparency of our experimental design. By comparing the results from a sparse reward (RQ2) and an enhanced reward (RQ3), the impact of our reward shaping is made clear.
> > >
> > > **3. On the Scope of Contribution (Request #5)**
> > >
> > > Finally, we would like to address your most critical point about the contribution feeling incomplete without downstream validation. Your feedback motivated us to conduct a new, crucial experiment with significant positive results.
> > >
> > > We are pleased to report that fine-tuning a baseline LLM with our GFlowNet-generated data led to a **dramatic performance increase from 0% to 28.6% accuracy** on a strict, trajectory-prediction task. This result provides direct, empirical proof of our framework's practical utility.
> > >
> > > **For a detailed breakdown of the experimental setup, qualitative and quantitative results, and efficiency analysis, please see our official public response titled "Author's Official Comment: New Downstream Experimental Results and Analysis."**
> > >
> > > We are confident that these substantial revisions and new experimental results fully address your concerns and make our paper a much more complete and impactful contribution. Thank you once again for your detailed and constructive feedback.

---

> > > > ### Comment · Reviewer_VQao · 2025-07-27
> > > >
> > > > Thanks a lot for the detailed clarifications and the additional experimental results. These have definitely strengthened the paper.

---

### Author Response · Authors · 2025-07-08
**Author's Official Comment: New Downstream Experimental Results and Analysis**

**To: All Reviewers**

We would like to express our sincere gratitude for your insightful feedback. The most critical and common point raised was the need for downstream validation to prove the practical utility of our GFlowNet-generated trajectories. We took this to heart and conducted a new experiment to address this concern directly.

Due to time constraints, these new results have not yet been incorporated into the current manuscript. However, we present them here in detail to demonstrate our commitment to addressing your feedback. **If the paper is accepted, we guarantee that all new results and analyses presented below will be included in the final camera-ready version.**

---

#### **1. Downstream Task: From Zero Comprehension to Actionable Reasoning**

Our experiment was designed to empirically validate whether our synthetic trajectories can enhance the reasoning capabilities of a large language model on ARC tasks.

**Experimental Setup and Results**

We fine-tuned a powerful baseline model, `Llama 3.1 8B Instruct`, with approximately 10k GFlowNet-generated trajectories per task. The model's objective was to solve 7 different ARC problems by predicting the **entire correct action sequence**, a much stricter criterion than simply matching the final output grid.

The results show a definitive qualitative leap in performance:

| **Task ID** | **Baseline (Llama 3.1)** | **Fine-tuned w/ GFlowNet Data** |
| :------------------- | :----------------------: | :-----------------------------: |
| 86 (25ff71a9)        |         ✗ (Fail)         |            ✗ (Fail)             |
| 139 (6150a2bd)       |         ✗ (Fail)         |            ✓ (Pass)             |
| 149 (6773b310)       |         ✗ (Fail)         |            ✗ (Fail)             |
| 154 (6855a6e4)       |         ✗ (Fail)         |            ✗ (Fail)             |
| 178 (74dd1130)       |         ✗ (Fail)         |            ✓ (Pass)             |
| 240 (9d9215db)       |         ✗ (Fail)         |            ✗ (Fail)             |
| 379 (ecdecbb3)       |         ✗ (Fail)         |            ✗ (Fail)             |
| **Overall Accuracy** |      **0/7 (0%)** |       **2/7 (28.6%)** |

**Qualitative and Quantitative Analysis**

The performance jump from 0% to 28.6% represents a significant qualitative leap. The baseline model was fundamentally unable to comprehend the ARC problem format (DSL), generating natural language analyses (e.g., *"Let's analyze the pattern..."*) instead of valid actions. In contrast, the fine-tuned model learned the DSL, generating syntactically valid action sequences (e.g., `[left_rotate, ..., submit]`) and successfully solving two problems that require geometric transformations. This demonstrates that our GFlowNet-generated data successfully taught the LLM an **actionable, step-by-step problem-solving logic.**

#### **2. Efficiency and Scalability of GFlowNet Augmentation**

Our framework is not only effective but also remarkably efficient, making it a highly practical tool.

| **Task ID** | **Time Required (to generate 100k)** | **Generation Rate (trajectories/hr)** |
| :------------------- | :----------------------------------: | :-----------------------------------: |
| 154 (6855a6e4)       |                27 min                |                222k/hr                |
| 178 (74dd1130)       |                46 min                |                130k/hr                |
| 149 (6773b310)       |                50 min                |                120k/hr                |
| 139 (6150a2bd)       |                52 min                |                115k/hr                |
| 86 (25ff71a9)        |                53 min                |                113k/hr                |
| 240 (9d9215db)       |                83 min                |                 72k/hr                |
| 379 (ecdecbb3)       |                84 min                |                 71k/hr                |
| **Average** |              **56.4 min** |               **120k/hr** |

Generating 100,000 trajectories takes, on average, less than an hour—a task that would require thousands of hours of human annotation. This immense efficiency and scalability are key advantages of our approach.

#### **3. Practical Significance and Future Directions**

While the overall accuracy of 28.6% shows there is room for improvement, these results serve as a strong proof-of-concept. They confirm that GFlowNet can be a practical data augmentation tool for complex reasoning problems like ARC-AGI. The limitations also provide clear future research directions, such as scaling the data volume and generating data for more diverse problem types (e.g., symmetry, pattern-filling).

We believe these new findings directly and thoroughly address the reviewers' primary concerns. We are grateful for the feedback that led us to this more robust and empirically validated result.

---

> ### Author Response · Authors · 2025-07-15
> **Announcing Additional Key Improvements in the Final Manuscript**
>
> **To the Reviewers,**
>
> Following our last response, we would like to inform you that we have conducted the following additional analyses and enhancements to further clarify the contributions of our work and improve the completeness of the manuscript. Your insightful feedback has played a crucial role in making our research more robust. The key improvements are as follows:
>
> **1. Formalization of Downstream LLM Experiment Results into a Dedicated Section (Most Significant Change)**
> We have incorporated the downstream LLM experiment results, previously shared only in our rebuttal, into a formal section of the manuscript. The new **Section 4.3, "Downstream Evaluation on Large Language Models" (page 22)**, details the experimental design, the quantitative results summarized in **Table 14** (an accuracy improvement from 0% to 28.6%), and the qualitative difference from the baseline model (i.e., its ability to comprehend the DSL), thereby demonstrating the practical utility of our framework .
>
> **2. Deepened Analysis of Efficiency and Scalability (New Appendix E)**
> We have added a new **Appendix E (page 40)** that provides an in-depth analysis of the data generation efficiency and scalability of our GFlowNet framework.
> * This appendix quantitatively presents the trajectory generation speed for each task in **Table 22**. In the manuscript, to emphasize the applicability at test-time for benchmarks like the ARC Prize, we have changed the unit to 'trajectories per minute' (averaging 2,000/min). We clarify that while the noted values differ from the 'per hour' figures in our previous response, they represent the same underlying performance.
> * Furthermore, **Figure 16** analyzes the logarithmic growth trend of trajectory diversity, clearly illustrating the practical cost-effectiveness trade-off offered by our methodology.
>
> **3. Highlighting the 'Test-Time Augmentability' Advantage**
> We have emphasized the importance of our methodology's key advantage, **'Test-Time Augmentability,'** throughout the manuscript. This is because the ability to generate diverse solutions on the fly for unseen problems, without any pre-existing trajectories, is a core differentiator of our approach. Specifically, we have specified this concept in multiple sections, including the **Abstract**, **Introduction (page 2)**, and **Methods (Section 3.1)**, to clarify how our research differs from existing data augmentation methods and why it is more suitable for generalization benchmarks like ARC.
>
> **4. Updating the List of Contributions**
> Reflecting these additional analyses and experimental results, we have added a fourth item, **'Test-Time Efficiency and Yield Analysis,'** to the **list of Contributions (page 3)** to better define the contributions of our research.
>
> ---
> **Other and Minor Corrections**
>
> Additionally, we have corrected a few minor issues that arose during the revision process.
> * **Table Numbering:** The table numbers in the appendix were naturally updated as new content was added to the main body. (e.g., Tables 18 and 19 mentioned in our response to Reviewer XpcR are now Tables 20 and 21 in the final version).
> * **Task ID Correction:** There was a misnotation error with some Task IDs in our previous response, but we have corrected this in the final manuscript to ensure data consistency.
>
> Once again, we express our deep gratitude for your insightful feedback. We are confident that these improvements have made our paper a more robust and impactful contribution.

---

### Decision · Action_Editor_7yBY · 2025-08-20

**Recommendation:** Accept with minor revision

**Audience:**

Yes

**Audience Explanation:**

This work will be of interest to the TMLR community. It advances the emerging line of research on using GFlowNets for structured reasoning tasks, introduces practical methods for trajectory-level augmentation, and provides empirical results demonstrating efficiency and utility.

**Claims And Evidence:**

Yes

**Claims Explanation:**

This paper proposes to use GFlowNet to synthesize diverse solution trajectories for ARC-AGI tasks. The paper introduces a geometric forward policy and human-inspired reward shaping to further improve performance. The method shows good performance for large-scale trajectory generation and demonstrates benefit when fine-tuning a language model.

The technical contributions are clearly described and supported by experiments. The framework is well-grounded in prior work on GFlowNets and reasoning augmentation, and the proposed modifications (reward design, geometric policy) are logically consistent and empirically validated. I found no technical flaws or inconsistencies in methodology, experiments, or analysis.

Several reviewers were especially concerned about how the scope is somewhat narrow (focused on ARC-AGI) and that claims about human-like reasoning may be overstated. The authors can consider expanding Section 4.3 (Downstream Evaluation on Large Language Models) to further validate this claim. To expand the section, authors can consider adding LLM post-training baselines and more tasks.